# COMPOSER: A SEARCH FRAMEWORK FOR HYBRID NEURAL ARCHITECTURE DESIGN

**Bilge Acun**[*], **Prasoon Sinha**[*,†]**, Newsha Ardalani, Sangmin Bae, Alicia Golden, Chien-Yu Lin, Meghana Madhyastha, Fei Sun, Neeraja J. Yadwadkar**[†]**, Carole-Jean Wu**

FAIR at Meta, [†] The University of Texas at Austin

[*] Joint first authors, work done at Meta

`acun@meta.com`, `prasoon.sinha@utexas.edu`

## ABSTRACT

Hybrid model architectures that combine computational primitives (e.g., Attention, MLP) in different ratios have shown promising performance beyond Transformers. Some studies have shown that different interleavings of primitives can affect model quality as well. However, prior works explore the hybrid model architecture design space manually. Due to the large design space and training costs, discovering hybrid models that combine key computational primitives for pretraining is challenging. In this work, we take a principled approach in designing a modular hybrid model architecture search framework — *Composer*. Composer explores model architectures at a small scale and extrapolates the top-performing model architectures to a larger scale using our proposed scaling strategies. Using Composer, we discover new hybrid LLM architectures that outperform Llama 3.2. Compared to Llama 3.2 and previous state-of-the-art baselines, the new model architectures consistently reduce validation loss at parameter scales of 350M-8B and improve evaluation accuracy on the downstream tasks by 2-2.1% on average while improving both training and inference efficiency.

## 1 INTRODUCTION

Transformers (Vaswani et al., 2017) have long served as the foundation of large language models (LLMs), powering mainstream models like BERT (Devlin et al., 2019) and GPT (Radford et al., 2019). The standard Transformer architecture features a fixed sequential interleaving of self-attention and multi-layer perceptron (MLP) layers. While this design remains effective, recent works demonstrate that *hybrid LLM architectures*—which deviate from the conventional Transformer stack—can further improve model quality. For example, unlike Transformer-based architectures which stack a 1:1 ratio of computational primitives, approaches like Qwen3-Next (Qwen, 2025), Mamba-2 (Dao & Gu, 2024), and MAD (Poli et al., 2024) adjust the ratio of Transformer and State Space Model (SSM) primitives within stackable blocks, skewing the composition toward one type or another. Meanwhile, other approaches rearrange the primitives in more sophisticated patterns by breaking the stacked structure. For example, DeepSeek's V3 MoE model (DeepSeek-AI, 2025) incorporates a few dense MLPs in the initial layers followed by sparsely activated MoEs. FastViT (Vasu et al., 2023) uses convolution in the beginning stages of the model and attention in the later stages. Sandwich Transformer (Press et al., 2020) reorders the interleaving of attention and MLP layers without changing the ratio.

Despite promising advances in hybrid LLM architectures, the model architecture design process is manual and based on intuition — no systematic framework exists today to enable automatic, efficient discovery of hybrid LLM architectures that perform well at scale. An effective search framework is essential given the vast model architecture design space–for example, a 32-layer hybrid LLM consisting of simply attention and MLP layers already yields over 4 billion ($2^{32}$) possible architecture configurations. The Nemotron model family builds a Post Neural Architecture Search (PostNAS) framework that prunes and replaces blocks of pre-trained models (Bercovich et al., 2025; Gu et al., 2025). STAR (Thomas et al., 2025) presents an initial attempt towards a framework targeting pre-training hybrid LLMs from scratch; however, its design assumes conducting search on the target dataset for edge use cases. We find that conducting search with web-scale datasets is either ineffective or impractical for performance evaluation.

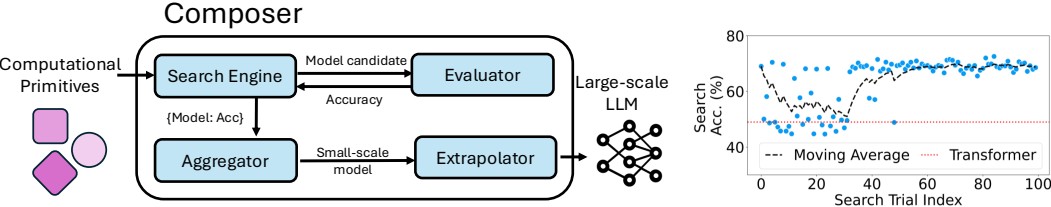

Figure 1: Design of the Composer search framework (left) and the accuracy convergence over search trials (right).

In this work, we take a principled approach to answer the key research question: how do we design a model architecture search framework that automatically and efficiently discovers novel hybrid LLM architectures for pretraining that outperform state-of-the-art models at scale? We find that when scaling down dataset and model size via Chinchilla scaling laws (Hoffmann et al., 2022), the model quality of small explored hybrid LLMs does not reflect large scale performance. Therefore, we require new methodologies for small-scale neural architecture search that is reflective of at-scale performance. We design *Composer*, a hybrid neural architecture search (HNAS) framework. Unlike traditional neural architecture search that assumes fixed interleavings of computational primitives when searching over model hyperparameters (e.g., model width, number of layers) (Wang et al., 2025; Javaheripi et al., 2022b), Composer rearranges the interleaving pattern and ratio of computational primitives to automatically discover high-performing hybrid LLM architectures. Leveraging our modular framework, we conduct an extensive exploration to answer the following key design questions:

**1. What is an effective search algorithm for conducting small-scale model architecture search?**
Searching the vast design space requires efficient search algorithms and effective techniques for scaling down model dimensions. Composer's *Search Engine* generates and searches through candidate hybrid LLM architectures of bounded size. We propose both Bayesian Optimization and iterative search techniques for efficient search, and provide key architectural design principles that outperform standard Transformer architectures.

**2. What datasets should we use to evaluate scaled-down hybrid LLM architectures?** For efficient search, candidate architectures need to be trained and evaluated efficiently with either small scale proxy datasets or sampled-down versions of target datasets. We investigate the efficacy of three different datasets for small-scale search with Composer's *Evaluator*.

**3. How do we synthesize the model candidates from search results into a final hybrid LLM?**
Small-scale search can produce multiple high performing model candidates. Composer's *Aggregator* synthesizes the search results into a final hybrid LLM using a clustering technique with the goal of generalizing to large-scale architectures.

**4. How do we extrapolate discovered small hybrid LLM architectures to large-scale model sizes?** Once the small model candidates are found, the architecture needs to be scaled. Composer's *Extrapolator* scales up the discovered architecture to an arbitrary desired size (we scale up ∼1000× and show results up to 8B) with two proposed techniques: stacking and stretching.

As our focus in this work is to build a HNAS framework in a principled manner, we design LLMs with Attention/MLP hybridization. After conducting our framework's design exploration, we demonstrate that the top-two performing hybrid LLMs discovered with Composer, termed *Composite architectures*, outperform Llama 3.2 and several other state-of-the-art model architectures. Our Composite architectures, consisting of advanced interleavings and a 1:2 ratio of Grouped Query Attention (GQA) to SwiGLU layers, reduce validation loss by 0.03 and increase accuracy on downstream LLM evaluation tasks by 2-2.1% on average. across a variety of model sizes and training budgets. Meanwhile, we increase training throughput by 1.25×, reduce KV cache size by 1.69×, and reduce inference latency by 1.33× on average.

## 2 COMPOSER DESIGN

We introduce *Composer*, an efficient and automatic hybrid neural architecture search framework, as illustrated in Figure 1. The framework ingests a set of computational primitives and uses Bayesian

Optimization to perform hybrid model architecture search at small scale (e.g., million parameter models) and discover architectures that will perform well when extrapolated to ~1000x scale (e.g., 3B model size). Composer has four core components: the HNAS Engine, Evaluator, Aggregator, and Extrapolator. After, Composer outputs a new hybrid LLM at a given size (e.g., 3B) ready for pre-training. We describe each of Composer's core components next.

## 2.1 Hybrid Neural Architecture Search Engine

The HNAS Engine systematically discovers high-quality, novel hybrid LLM architectures. We define a hybrid LLM as a sequence of computational primitives (e.g., Attention, MLP) from the set $\mathcal{P} = \{p_1, p_2, \ldots, p_Z\}$, where $Z$ is the number of unique computational primitives. For a fixed number of layers $N$, a hybrid LLM is formally defined as $\mathbf{a} = (a_1, a_2, \ldots, a_N) \in \mathcal{P}^N$, where $a_i \in \mathcal{P}$ specifies the primitive at layer $i$. The discrete search space contains all possible primitive arrangements $\mathcal{A}_N = \{(a_1, \ldots, a_N) : a_i \in \mathcal{P}, \forall i \in [N]\}$, yielding $|\mathcal{A}_N| = Z^N$ candidate architectures.

The discrete search space exponentially grows with the target model size, since the depth of the target size $N$ grows with model size (e.g., Llama 3.2's depth increases from 32 to 72 layers from 1B to 8B model size). To efficiently navigate the large design space, we propose three search methodologies.

**(1) One-Shot Search:** This methodology performs a one-shot $n$-layer search, where $n \leq N$. If $n < N$, the discovered model is then extrapolated, or scaled up, to target size via one of our proposed extrapolation techniques in § 2.4. We leverage Bayesian Optimization with Gaussian Process surrogate modeling to navigate through the hybrid architecture design space of size $Z^n$. We build upon the Ax framework (Olson et al., 2025; Meta Platforms, 2024), using a BoTorch SingleTaskGP model with an RBF Kernel, dimension-scaled priors, and qLogNEI acquisition function for single objective optimization. Bayesian Optimization offers better sample efficiency and uncertainty modeling compared to Reinforcement Learning or evolutionary search. The optimizer targets a black-box function $f : \mathcal{A}_n \to \mathbb{R}$ measuring validation accuracy after pre-training: $f(\mathbf{a}) = \textit{Accuracy}(\textit{PreTrain}(\mathbf{a}, \mathcal{D}_{\text{train}}), \mathcal{D}_{\text{val}})$, where $\mathcal{D}_{\text{train}}$ and $\mathcal{D}_{\text{val}}$ are the training and validation dataset used by the HNAS Evaluator (§ 2.2). We solve the optimization problem: $\mathbf{a}^* = \arg\max_{\mathbf{a} \in \mathcal{A}_n} f(\mathbf{a})$.

**(2) End-Layer Incremental Search:** This methodology prunes the design space and incrementally builds a hybrid LLM to target size. Specifically, we perform $n$-layer iterative search, starting with $n$ layers, where $n \mid N$ and $n > 1$. Then, we progressively increase the architecture depth by $n$ layers at each step, fixing computational primitives for the previous layers and searching only the last $n$. Hence, at any given iteration, there are only $2^n$ unique architectures to evaluate. This process repeats up to $N$ layers with a total of $N/n$ iterations.

**(3) Middle-Layer Incremental Search:** This methodology follows a similar process to (2), but searches over middle rather than end layers. The Search Engine first conducts an $n$-layer search. Then, for each subsequent $n$-layer expansion (e.g., $n$ to $2n$, $2n$ to $3n$), we fix the beginning and end layers by splitting the previously found architecture down the middle. We then only search over the middle $n$ layers. This approach ensures that at each stage, the central layers are optimized while the outer layers remain fixed from the prior iteration. This process repeats up to $N$ layers.

**Further improving search efficiency via width scaling.** To reduce search cost, the one-shot search method searches over fewer layers than the target model size (i.e. $n << N$) while the incremental search techniques prune the design space with their iterative nature. However, without reducing the width of computational primitives, the search cost remains prohibitive (§ 3.3). Hence, we also scale down the widths of the primitives compared to the target model size. We find that this not only reduces search cost, but also enables Composer to discover higher quality hybrid LLMs (§ 3.3).

## 2.2 Hybrid Neural Architecture Evaluator

During search, the HNAS Evaluator trains and evaluates candidate hybrid LLMs with a small dataset to provide fast, reliable signals on the potential quality of the architecture at scale. The dataset used during search is crucial for accurately identifying architectures that perform well at scale when pre-trained with the target dataset; we use DCLM (Li et al., 2024), a large web-scale text dataset, for pre-training. We empirically evaluate three datasets for small-scale search in § 3.2: randomly sampled-down DCLM (Li et al., 2024), MAD (Poli et al., 2024) which is a synthetic token-manipulation

| Composer Component | Default Methodology | Details |
|---|---|---|
| HNAS Search Engine | One-Shot Search | Both 6 and 16-layer search |
| HNAS Evaluator | MAD's synthetic tasks | Details in Appendix E.2 |
| HNAS Aggregator | $N_0$ clustering | K-means with 5 clusters |
| HNAS Extrapolator | Stacking and stretching | Stack 6-layer search, stretch 16-layer search |

Table 1: Default methodology used during ablation study of each of Composer's components.

dataset designed to probe different capabilities of LLMs, and finally BabiStories, which is a synthetically generated children's story dataset (Zhang et al., 2025) (an OSS model generated version of the TinyStories dataset (Li & Eldan, 2024)). Appendix E.2 provides further details about each dataset.

## 2.3 HYBRID NEURAL ARCHITECTURE AGGREGATOR

The Aggregator post-processes search results, either after the search process completes with One-Shot Search or after each iteration of End-Layer/Middle-Layer Incremental Search, to finalize the small-scale hybrid LLM. We propose $N_c$ clustering to select the primitive at each layer conditioned on the sequence of the $c$ previously selected primitives. Formally, let $\mathcal{C} = \{\mathbf{a}^{(1)}, \mathbf{a}^{(2)}, \dots, \mathbf{a}^{(T)}\}$ denote the set of top candidate architectures, where each architecture $\mathbf{a}^{(t)} = (a_1^{(t)}, a_2^{(t)}, \dots, a_n^{(t)})$ is a sequence of computational blocks indexed by layer. We use K-means clustering, with validation accuracy of the candidate models from search as the target metric, to populate $\mathcal{C}$. For a given layer $i$, let $\hat{\mathbf{a}}_{i-c:i-1}$ denote the selected sequence of blocks at the previous $c$ layers. Then,

$$N_c: \quad \hat{a}_i = mode\Big(\{a_i^{(m)} \mid \mathbf{a}^{(m)} \in \mathcal{C}, \ a_{i-c:i-1}^{(m)} = \hat{\mathbf{a}}_{i-c:i-1}\}\Big) \quad \forall i \in [1, n] \tag{1}$$

where $\hat{a}_i$ denotes the selected block at layer $i$, and mode($\cdot$) returns the most frequent computational primitive among the filtered candidate architectures that match the selected prefix.

When $c = 0$, $N_0$ clustering selects the dominant block at each layer among the top candidate architectures independently without conditioning on prior layers. When $c = 1$, $N_1$ clustering conditions the block choice at each layer on the block selected at the immediate preceding layer. Finally, for $c = i - 1$, the block selected at each layer index $i$ is conditioned on the entire sequence of previously selected blocks, enforcing full prefix consistency. We also consider simply using the best architecture discovered during the search process. We find that $N_0$ clustering ($c = 0$) produces the best hybrid LLM at scale; clustering over all top-performing candidate LLMs smoothes out noise or overfitting that may occur during small-scale search. We include this analysis in Appendix D.5.

## 2.4 HYBRID NEURAL ARCHITECTURE EXTRAPOLATOR

With a final small hybrid LLM, the HNAS Extrapolator scales up the architecture to the desired model size (e.g., 3B). If the depth of the hybrid LLM $n$ matches the depth of target model size $N$, the Extrapolator simply scales the width back up to that of Llama 3.2. Otherwise, the Extrapolator also scales up the depth using one of two techniques, stretching or stacking. We describe each next.

**Extrapolation via stretching.** Stretching scales up the depth of the hybrid LLM while keeping the interleaving pattern and ratio of computational primitives the same as the discovered, small hybrid LLM, thereby "stretching" the architecture. Formally, we partition a hybrid architecture $\mathbf{a}$ into $G$ contiguous groups, where each group $g$ contains identical primitives $p$:

$$\mathbf{h} = \big[(p_1, h_1), (p_2, h_2), \dots, (p_G, h_G)\big], \quad \text{where} \quad p_g \in \mathcal{P}, \quad h_g \in \mathbb{Z}^+, \quad \sum_{g=1}^{G} h_g = n. \tag{2}$$

Given the desired and current model sizes $M$ and $m$, respectively, we define a scaling factor $s = \frac{M}{m}$. The scaled architecture is then

$$\mathbf{h}^{\text{large}} = \big[(p_1, \lceil s \cdot h_1 \rceil), (p_2, \lceil s \cdot h_2 \rceil), \dots, (p_G, \lceil s \cdot h_G \rceil)\big]. \tag{3}$$

Intuitively, the scaling factor represents the ratio by which the model size is increased, scaling the number of layers in each group proportionally to reach the target model size. Appendix **??** provides a visualization of stretching.

**Extrapolation via stacking.** This technique considers the discovered small hybrid LLM as a stackable block. To scale up to the desired parameter size, we stack $s$ copies of this block sequentially, where $s$ is the scale factor $s = \lfloor \frac{M}{m} \rfloor$. We also define a remainder scaling term $r = M \bmod (m \times s)$. We use this term to shrink the discovered LLM following Equation 3 and append these layers to the end of the stacked architecture. This ensures the final architecture closely matches the target size.

Figure 2 depicts how each extrapolation methodology works.

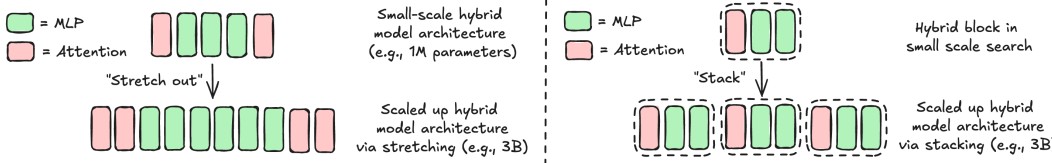

Figure 2: Illustration of extrapolating small-scale hybrid architecture to large-scale via stretching (left) or stacking (right).

## 3 DESIGN EXPLORATION OF COMPOSER'S CORE COMPONENTS

We perform a design exploration study with different methodologies for each of Composer's components proposed in § 2. For each component under test, we fix the methodology of the other components (Table 1) based on our findings of which methodology performs best. Each methodology produces different small hybrid LLMs. To evaluate each methodology's efficacy, we scale up the hybrid LLMs to 1B parameters, pre-train them with the DCLM dataset (Li et al., 2024), and compare validation loss. We use the same width for all LLMs, ensuring performance differences arise due to differences in the ratio and interleaving of computational primitives between hybrid LLMs. We set width dimensions to that of Llama 3.2 1B (2048×8192, 32 attention heads, 8 KV heads). Appendix B details the architectures produced by each methodology. We provide further details of our pre-training setup in Appendix C.3.

### 3.1 EXPLORATION OF SEARCH METHODOLOGIES

We first evaluate the model quality versus search cost of the three search methodologies we propose in § 2.1. For One-Shot Search, we evaluate two variants with $n = 6$ and $n = 16$, performing 6-layer and 16-layer search, respectively. For End-Layer and Middle-Layer Increment Search, we set $n = 2$ and $n = 4$ to perform 2-layer and 4-layer incremental searches, respectively, up to N=32 (Llama 3.2 1B, the target model size, has 16 Transformer blocks or 32 layers). Figure 3-left presents the at-scale DCLM validation loss and search cost for each methodology.

> **Observation 1:** All three proposed search methodologies produce hybrid LLMs that outperform Llama 3.2 across compute budgets, showing that breaking the standard Transformer architecture with more advanced interleavings of attention and MLP layers improves model quality.

The End-Layer Iterative search methodology produces a hybrid LLM with (roughly) a 1:1 Attention-to-MLP ratio, like Llama 3.2. However, during search, Composer learns that breaking the standard sequential Transformer interleaving for more intelligent arrangements with variable sized groups of Attention and MLPs interleaved can greatly improve model quality.

> **Observation 2:** In addition to the ordering of the layer types, we find that a 1:2 Attention-to-MLP ratio can further improve model quality compared to a 1:1 ratio.

Both Middle-Incremental and One-Shot Search outperform End-Incremental Search (0.01-0.02 reduction in validation loss across training budgets). Only 40% of Middle-Incremental's hybrid LLM consists of attention layers. Both One-Shot searches further reduce the number of attention layers to only 33% of the hybrid LLM's depth while maintaining model quality with lower search costs compared to End-Incremental (1.4-2.1× reduction in search cost). Hence, despite all three proposed search methodologies producing high quality LLMs compared to Llama, we leverage One-Shot

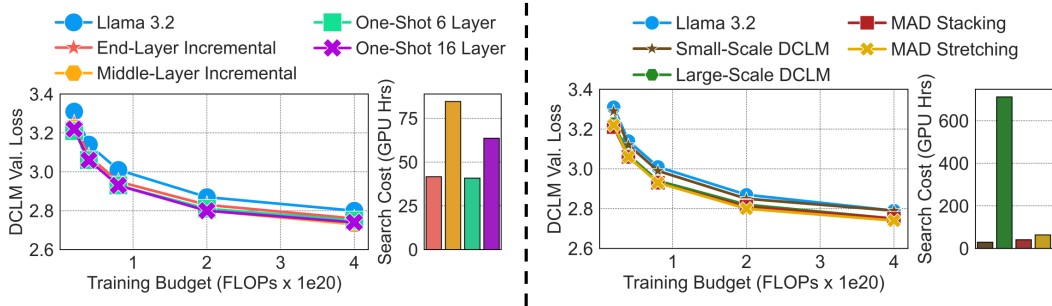

Figure 3: Design exploration of (1) search methodology for Composer's HNAS Search Engine (left side) and (2) datasets for Composer's HNAS Evaluator (right side). We report the model quality at 1B scale and search cost for each technique.

search throughout the rest of this paper as the methodology for Composer's Search Engine due to better model quality and search cost trade-off.

## 3.2 EXPLORATION OF EVALUATION METHODOLOGY

It is crucial that the dataset used for small-scale search enables Composer to identify representative hybrid LLMs that perform at scale. However, searching over target model sizes (e.g., 3B) with web-scale datasets is prohibitive. As DCLM is our target large-scale pre-training dataset (Li et al., 2024), we evaluate two approaches for leveraging it during search: (1) scaling down both the model and the dataset via scaling laws, (2) scaling down only the dataset while keeping the model large.

> **Observation 3**: Compared to using the traditional scaling laws methodology to scale down web-scale datasets and model sizes, we find small-scale proxy datasets can be more effective in guiding the search process.

We reduce the model and data size, following Chinchilla scaling laws (Hoffmann et al., 2022), down to a 4-layer 4M parameter model trained on 98.6M tokens (12K samples). After the 4-layer search, we extrapolate to 1B size by stacking. Figure 3-right shows that this methodology, *Small-Scale DCLM*, only slightly outperforms Llama 3.2 and improvements diminish with larger training budgets (4e20 FLOPs). Therefore, as an alternative approach, we sample down only the dataset size to 12K samples while keeping model size large. We perform a 16-layer search and extrapolate to 1B size via stretching. Figure 3-right shows that this methodology, labeled *Large-Scale DCLM*, yields high quality models that consistently outperform Llama 3.2; however, the search cost is large (>25 GPU days). Ultimately, we find that DCLM is either ineffective or impractical for performance evaluation.

We also evaluate the efficacy of small-scale synthetic datasets as the evaluation mechanism: MAD (Poli et al., 2024) and BabiStories (Zhang et al., 2025) (Appendix D.6 details our experience with BabiStories). With MAD, we conducted 6-layer and 16-layer searches, extrapolating the produced LLMs to 1B parameters via stacking or stretching, respectively. MAD reduced search cost by > 8× compared to *Large-Scale DCLM* while producing hybrid LLMs that consistently outperform Llama 3.2 at scale. We suspect this for two reasons: its token-manipulation tasks are (1) learnable by small models since they have a small vocabulary size, and (2) representative of large-scale LLM tasks. This enables efficient small-scale search with strong large-scale performance. While further study of the proxy dataset quality is required, Composer's Evaluator uses MAD throughout this work.

## 3.3 EXPLORATION OF EXTRAPOLATION AND SCALING METHODOLOGIES

We study the efficacy of stacking versus stretching to extrapolate small hybrid LLMs to larger sizes. We perform search with varying the number of layers $n$ Composer searches over from 4 to 32. We then stretch and stack the discovered hybrid LLMs to 1B scale, and pre-train them with DCLM for 4e20 FLOPs. Figure 4-top reports DCLM validation loss for each model variant.

**Observation 4**: Stacking is an extrapolation mechanism that consistently produces high-quality hybrid LLMs across different $n$-layer searches. However, stretching hybrid LLMs discovered from larger $n$-layer search configurations enables Composer to discover more creative interleavings of computational primitives, resulting in higher quality hybrid LLMs.

Generally, stacking as a mechanism for extrapolation works well regardless of the number of layers searched. Stretching does not work well when conducting search over a small number of layers (e.g., 2A + 4M), as it creates hybrid LLMs where Attention dominates beginning layers and MLP dominates end layers without any good interleaving pattern in the middle (e.g., 10A + 20M at scale). However, an inflection point occurs where stretching consistently outperforms stacking beyond 16-layer searches. Expanding the search space with 16-layers allows Composer to creatively find new interleaving patterns that it cannot explore with the small, restricted search space from 6-layer search. Moreover, stretching preserves information by maintaining transitions from one computational primitive to another, enabling the hybrid architecture to better capture global information (i.e., propagate signals and gradients across transition points to capture more complex dependencies). However, conducting search beyond 16 layers produces lower quality hybrid LLMs; we hypothesize this is due to larger number of layers exponentially increasing the search space, making it challenging for Composer to discover higher quality models within the fixed trial budget used for search. Per Figure 4, we choose two different depths to perform search with: stack 6-layer searches and stretch 16-layer searches.

**Observation 5:** Scaling down both model width and depth greatly reduces search cost by $>6\times$ while also discovering hybrid LLMs that perform better at scale. This is because conducting small-scale search with similar width-to-depth ratios as target model sizes preserves model characteristics during search and, thus, increases the likelihood of high model quality at the target scale.

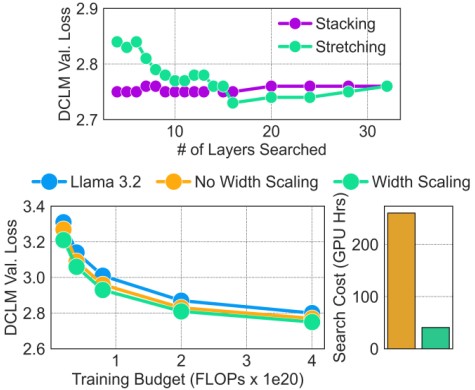

Figure 4: [Top] Model quality of different hybrid LLMs produced after stacking and stretching with various search depths. [Bottom] Model quality and search cost with and without width scaling.

Figure 4-top shows the importance of scaling down depth from the target model size to reduce the design space and discover quality hybrid LLMs (e.g., 32 versus 16 layers stretching). We also study the importance of scaling down width. Figure 4-bottom compares the model quality and search cost of 6-layer search with width scaling (128x258, 16 heads) and without width scaling (Llama 3.2 1B width).

Scaling down the width reduces search cost by $6.38\times$. Without width scaling, Composer conducts search with model sizes 100M-200M parameters. This increases the time to train/evaluate each candidate hybrid model, inflating the total search cost. Moreover, scaling down width enables Composer to produce higher quality models (validation loss reduces by 0.02-0.04). Without scaling down the width, Composer's candidate hybrid architectures during search have greatly skewed width to depth ratios: too wide and shallow. Hence, the hybrid architectures with the best performance at small scale do not translate to the best performance at scale. For example, without width scaling, Composer finds that the architecture 3A + 3M (3 Attention followed by 3 MLPs) performs best at small scale. However, when also scaling down width, Composer finds a better architecture with a 1:2 Attention-to-MLP ratio (2A + 4M) that outperforms the 1:1 ratio model at large scale.

## 4 EVALUATION

With finalized methodologies (Table 1), we conduct a 6-layer and 16-layer search and discover two unique hybrid LLMs:

$$\text{6-Layer Search Hybrid LLM} = 2A + 4M \qquad (4)$$

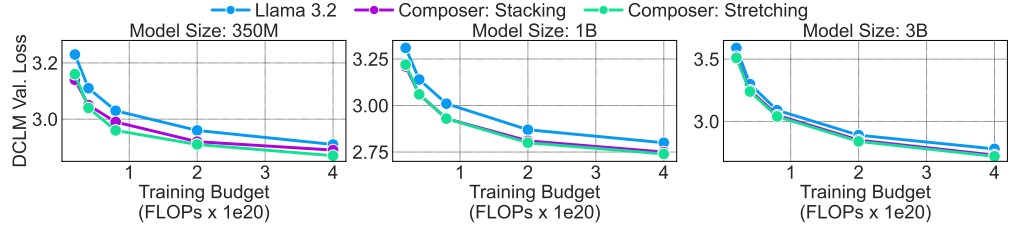

Figure 5: Validation loss across model sizes (350M-3B) and training budgets (2e19-4e20 FLOPs).

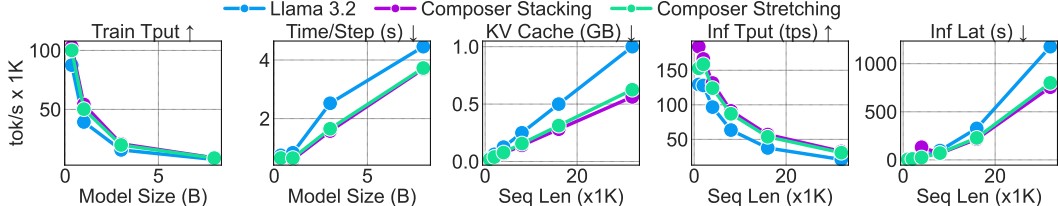

Figure 6: Composite LLMs' training and efficiency improvements compared to Llama 3.2.

$$\text{16-Layer Search Hybrid LLM} = 2A + 5M + 2A + 3M + 1A + 3M \tag{5}$$

We extrapolate these architectures to various sizes via stacking and stretching, respectively. We refer to any variant of Equation 4 and Equation 5 as Stacked and Stretched Composite LLMs, respectively.

## 4.1 IsoFLOP Analysis And Evaluation on Downstream Tasks

We evaluate our Composite LLMs against Llama 3.2 across a wide range of model sizes (350M-8B) and training budgets (2e19 to 4e20 FLOPs) to show our hybrid LLMs maintain predictive performance as scale increases. Figure 5 presents DCLM validation loss across model sizes and training budgets. Appendix D.4 reports performance across popular downstream tasks. We also include a scaling analysis of our Composite models in Appendix D.1.

> **Key Result 1:** Composite models are robust across model sizes, training budgets, and downstream tasks. They consistently reduce loss over Llama 3.2 by 0.05-1.0 and outperform Llama 3.2 on all the downstream tasks with performance improvements up to 2.8-8.3% (1.1-3.1% avg stacked and stretched).

## 4.2 Training and Inference Efficiency Evaluation

Figure 6 analyzes the training and inference efficiency of Composite LLMs. We include inference efficiency metrics at 1B scale with batch size 1; Appendix D.3 details inference data across model sizes and batch sizes. Composite models have fewer layers than Llama 3.2: stacking has 27 layers and stretching has 29 layers, while Llama 3.2 has 32 layers (16 Transformer blocks). Moreover, Composite models have a 1:2 Attention-to-MLP ratio (9-10 Attention layers), unlike the 1:1 ratio with Transformer architectures (16 Attention layers). These two properties greatly improve the efficiency of Composite models compared to Transformer based LLMs.

> **Key Result 2:** Composite models consistently improve training and inference efficiency. Compared to Llama 3.2, across model sizes, we increase training throughput by $1.25\times$, reducing per-step training time by $1.32\times$ on average. Across sequence lengths, at 1B scale, we improve inference latency by $1.33\times$ on average. Moreover, as a byproduct of having fewer Attention layers, we also reduce KV cache size by $1.69\times$.

## 4.3 COMPARISON AGAINST STATE-OF-THE-ART PREVIOUS WORKS

Next, we compare our hybrid architectures against three previous state-of-the-art works at 1B size: Sandwich Transformer (Press et al., 2020), Striped Attention (Poli et al., 2024), and the best performing architecture from STAR (Thomas et al., 2025). Sandwich Transformer has a 1:1 Attention-to-MLP ratio with a rearranged "sandwich" interleaving pattern: 8 Attention, followed by 8 MLP and Attention sequentially interleaved, ending with 8 MLP. The Striped Attention models apply a 1:2, 1:4, or 1:8 ratio of Attention to MLP layers, stacking multiple blocks of 1A + 2M (or 4M/8M) to desired sizes. STAR's best model consists of a mix of gated convolution, MLP, self-attention, and recurrent layers. Further architectural details are provided in Appendix B.6.

| Model | Loss ↓ | Arc C. ↑ | Hella. ↑ | Wino. ↑ | SciQ ↑ | PIQA ↑ | Arc E. ↑ | Avg. ↑ |
|---|---|---|---|---|---|---|---|---|
| Llama 3.2 | 2.80 | 29.8 | 53.1 | 55.8 | 80.6 | 71.8 | 61.03 | 58.69 |
| Sand. Transformer | 2.77 | 30.8 | 54.93 | 55.25 | 83.4 | 71.5 | 63.43 | 59.88 |
| 1:2 Striped Attn. | 2.81 | 29.0 | 52.9 | **56.4** | 80.0 | 72.6 | 62.92 | 58.97 |
| 1:4 Striped Attn. | 2.82 | 29.8 | 51.9 | 53.8 | 78.3 | 71.9 | 63.09 | 58.13 |
| 1:8 Striped Attn. | 2.85 | 30.7 | 50.9 | 52.3 | 75.7 | 71.9 | 62.58 | 57.35 |
| STAR* | - | 27.9 | 52.6 | 53.9 | 87 | 71.8 | 60.8 | 59 |
| Composite: Stacked | **2.77** | 28.84 | 54.56 | 55.72 | 87.6 | **73.56** | **64.73** | **60.83** |
| Composite: Stretched | **2.77** | **32.25** | **54.96** | 53.9 | **87.9** | 72.3 | 63.26 | **60.76** |

Table 2: Comparison of Composer's stacked and stretched hybrid architectures to LLM architectures from previous works: Sandwich Transformer (Press et al., 2020), Striped Attention (Poli et al., 2024), and STAR (Thomas et al., 2025) at 1B scale. Note that for Wino. and Arc E., accuracy is reported. For other tasks, normalized accuracy is reported.

We pre-train all LLMs with DCLM, except STAR, which could not be pre-trained since the hybrid model is not open-sourced. To make as fair of a comparison with STAR, we match its pre-training setup by training all models with 37.5B tokens (see Appendix C.5 for details). Table 2 reports each model's performance on the same downstream tasks as STAR, with STAR's results (which is pre-trained on a different dataset) taken directly from their paper.

> **Key Result 3:** Composite LLMs outperform state-of-the-art hybrid LLMs when trained with fixed number of tokens, reducing loss by 0.03 while increasing accuracy across the downstream tasks by up to 3.7% (1-2% avg).

## 4.4 ROBUSTNESS OF COMPOSER'S SEARCH FRAMEWORK

**(1) Comparison against randomly generated hybrid LLMs.**

We show Composer's robustness by comparing Composite LLMs against five randomly generated 16-layer hybrid LLMs stretched to 1B scale. Appendix B.7 details each architecture. The Composite LLM consistently outperforms the randomly generated architectures across training budgets (Figure 7-top). Those beginning with MLP layers (R1/R3) perform poorly with larger budgets. The remaining three architectures perform relatively well, but heavily skew with Attention layers. Composer's 1:2 Attention-to-MLP ratio provides the best model quality while also improving training and inference efficiency (§ 4.2).

**(2) Comparison of relative rankings at small and large scale.** We compare the relative rankings of candidate architectures evaluated during small-scale search and large-scale pre-training to show Composer's robustness. We rank the candidate LLMs after search and pre-train the p0-p100 LLMs at 1B

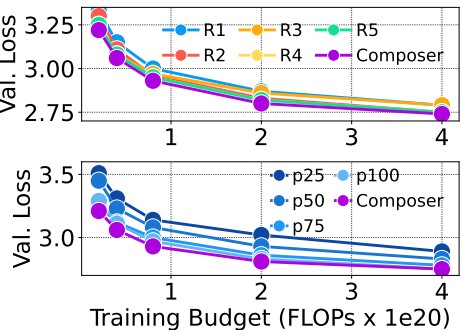

Figure 7: [Top] Model quality of the Composite LLM versus five randomly generated ones at 1B scale. [Bottom] Performance of ranked candidate architectures during search (p25-p100) at 1B scale. Relative rankings between small and large scale hold.

size. Figure 7 shows each architecture's validation loss after pre-training. Appendix B.8 details each architecture.

The Spearman rank correlation between small-scale 6-layer search and 1B scale is 0.97 on average across compute budgets, indicating high positive correlation (nearly identical rankings). The p0 (not shown, loss is >5) and p25 LLMs perform poorly: they consist of only MLP or only Attention layers, respectively. The p50 and p100 LLMs are all Attention-heavy (2:1 Attention-to-MLP ratio); these architectures do not perform as well as the best stacked architecture Composer produces with a 1:2 ratio. The p75 has a 1:2 Attention-to-MLP ratio, however, the layer interleavings are sub-optimal: the model begins with MLP layers and ends with Attention. Beginning with multiple Attention layers enables deep contextual understanding and feature extraction, while ending with MLP refines and projects features into accurate outputs. Composer's LLM after $N_0$ clustering adheres to both properties and provides superior model quality, demonstrating the advantage of $N_0$ clustering in smoothing out noise or overfitting that may occur during small-scale search.

## 5  DISCUSSION

Our study demonstrates Composer's efficacy in producing high-quality hybrid LLMs that achieve strong performance at scale on standard natural language understanding downstream tasks, such as PIQA (Bisk et al., 2020), WinoGrande (Sakaguchi et al., 2021), etc. Further work remains to understand Composer's efficacy for settings that require extended contexts or complex reasoning.

For example, if Composite LLMs outperforms Llama on long-context or reasoning tasks, and how our HNAS framework should be augmented to target these specific types of tasks needs further study. Specifically, we may require new small scale long-context or reasoning-specific datasets to effectively probe the performance of small hybrid LLMs for these tasks at large scale. However, as we find that sampled-down web-scale datasets are either impractical or ineffective for efficient small-scale search, we may require new token-manipulation datasets, like MAD (Poli et al., 2024), specifically targeting long-context or reasoning tasks. Ultimately, determining the best approach for dataset design and model evaluation in these domains remains an open question and a promising direction for future research.

## 6  CONCLUSION

We present Composer, a search framework that systematically discovers novel hybrid neural architectures that improve model quality at scale. We propose and evaluate the efficacy of several methodologies for each step in Composer's flow to build the framework in a principled manner. Specifically, we study different search, evaluation, aggregation, and extrapolation techniques, to efficiently conduct small-scale search and produce hybrid model architectures that continue to perform well at $1000\times$ larger than their searched size. Composite architectures outperform several state-of-the-art hybrid LLM architectures and Llama 3.2 in validation loss, downstream benchmarks, and model efficiency. Composer's efficient and extensible framework opens up possibilities to incorporate different computational primitives–such as Gated Delta Net, Mamba, Sliding Window Attention– in the search process and uncover new high-performance model architecture designs.

### ACKNOWLEDGMENTS

We thank Samuel Hsia, Michael Kuchnik, and Geet Sethi from the FAIR SysML team for their valuable discussions and feedback on this work.

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

## A RELATED WORK

**Hybrid LLM Architectures.** Hybrid LLM architectures have become increasingly popular to improve the quality and efficiency of LLMs. Several hybrid LLMs build new stackable blocks (analogous to a Transformer block (Vaswani et al., 2017)) by adding new computational primitives and adjusting the ratio of primitives within a block. For example, Qwen3-Next (Qwen, 2025), Mamba-2 (Dao & Gu, 2024), Jamba (Lieber et al., 2024) and MAD (Poli et al., 2024) all adjust the ratio of Transformer and State Space Model (SSM) primitives within a block, skewing the composition to a single computational primitive. Other hybrid LLMs consist of more sophisticated interleavings of computational primitives by breaking the conventional stacking-based structure. Command-A (Cohere et al., 2025) introduces hybridization by interleaving sliding-window and full attention layers in 3:1 ratio. Similarly, Llama-4 models (Meta, 2025) use interleaved attention layers without positional embeddings. DeepSeek-V3 671B model (DeepSeek-AI, 2025) incorporates three dense MLPs in its initial layers followed by sparsely activated MoEs. FastViT (Vasu et al., 2023) leverages convolutions in the beginning layers and attention in later stages. Sandwich Transformer (Press et al., 2020) maintains a 1:1 Attention-to-MLP ratio, but reorders the interleavings of the primitives.

**Neural Architecture Search Frameworks.** Neural Architecture Search more generally encompasses the art of automating the time consuming process of neural architecture design for a given task. There is a wealth of literature that studies how to (1) design an effective search space for a task (2) search through the space efficiently and (3) estimate the performance of a given architecture quickly. A popular class of search spaces are cell-based search spaces (White et al., 2023). In cell based search spaces, the search is performed over small cells and stacked several times in sequence to form an architecture (Zoph et al., 2018). Bayesian Optimization, reinforcement learning, and evolutionary search are classes of search strategies that have been shown to perform better than random search. A performance predictor is defined as any function which predicts the accuracy or relative accuracy of architectures, without fully training them. Various kinds of proxies exist including: learning curve extrapolation, zero-cost proxies and subset selection methods. NAS has been studied extensively for older and smaller models including CNN (ResNet style) models, NLP, and ASR models (Pham et al., 2018; Liu et al., 2018; Howard et al., 2019; Li et al., 2021; Tan & Le, 2021; Mecharbat et al., 2023). There have been various benchmarks released,including NAS-Bench-101 (Ying et al., 2019), NAS-BERT (Xu et al., 2021), NAS-Bench-ASR (Mehrotra et al., 2020), NAS-Bench-NLP (Klyuchnikov et al., 2022), to make NAS more reproducible. However, NAS has not been as well studied for modern Large Language Models (LLMs). One of the reasons is that the scale of the models makes it more difficult to design effective performance proxies that extrapolate to large scale. Some attempts include AutoMOE (Jawahar et al., 2022) that uses supernet training combined with evolutionary search to design an MoE architecture for machine translation and LiteTransformerSearch (Javaheripi et al., 2022a) that uses a zero cost performance proxy (the number of model parameters). However, the models searched over are smaller and older (OPT) models. More importantly, these traditional neural architecture search frameworks assume fixed interleavings/ratios of computational primitives when searching over model hyperparameters such as model width, number of layers, attention heads, output dimensions. Such zero-cost performance proxies may not apply when searching on a different model architecture variants.

**Neural Architecture Search Frameworks for Hybrid Models.** There are only few recent works that consider the hybrid model architecture design space in their search. The Nemotron model family builds a Post Neural Architecture Search (PostNAS) framework that prunes some of the modeling blocks or replaces Global-Attention blocks of pre-trained models with efficient Attention variants (Bercovich et al., 2025; Gu et al., 2025). This optimization in a post-training setting targets a completely different problem than designing new hybrid architecture for pre-training and, it does not reduce the large pre-training and experimentation costs of LLMs. STAR (Thomas et al., 2025) presents an initial attempt towards a framework targeting pre-training hybrid LLMs from scratch, however, its design assumes conducting search on the target dataset for edge use cases. We find that conducting search on web-scale datasets for LLMs is either ineffective or impractical for performance evaluation.

## B    Hybrid LLM Architecture Details

With our design exploration of Composer's components in § 3 and evaluation against Llama 3.2 and other state-of-the-art previous works, we evaluate hundreds of unique LLM architecture designs. In this section, we provide architectural details for each evaluated LLM.

Any LLM produced by Composer consists of Grouped-Query Attention with rotary positional embeddings (RoPE) (Su et al., 2023) (referred to as Attention or A for brevity) as the sequence mixing layer and SwiGLU (Shazeer, 2020) (referred to as MLP or M) as the channel mixing layer. We use RMSNorm (Zhang & Sennrich, 2019) for normalization and no linear bias term. All models tie the embedding layers.

| Model Size | Dim | Hidden Dim | Num Heads | Num KV Heads |
|---|---|---|---|---|
| 350M | 1536 | 4096 | 24 | 8 |
| 1B | 2048 | 8192 | 32 | 8 |
| 3B | 3072 | 8192 | 24 | 8 |
| 8B | 4096 | 14336 | 32 | 8 |

Table 3: Model dimensions for stretched and stacked Composite architectures, along with Llama 3.2 which we pre-train from scratch.

Table 3 describes the width dimensions of our models under different sizes. Below, we detail each architecture. We organize the descriptions section by section following § 3 and § 4 in our paper.

### B.1    Search Methodology Exploration: Hybrid LLM Architectures

In § 3.1, we explore the efficacy of three search methodologies: One-Shot Search, End-Layer Incremental Search, and Middle-Layer Incremental Search. We describe the Composite architecture produced by each search methodology before and after extrapolation to 1B scale.

**One-Shot 6-Layer Search**

- Small scale (400K parameters): 2A + 4M
- Large scale (1B parameters): 4×(2A+4M) + 1A + 2M

**One-Shot 16-Layer Search**

- Small scale (1.1M parameters): 2A + 5M + 2A + 3M + 1A + 3M
- Large scale (1B parameters): 4A + 9M + 4A + 5M + 2A + 5M

**End-Layer Incremental Search**

- Small scale (1M parameters): 2A + 2M + 2A + 2M + 4A + 1M + 3A + 2M + 1A + 1M + 1A + 6M + 1A + 3M + 1A
- Large scale (1B parameters): 2A + 2M + 2A + 2M + 4A + 1M + 3A + 2M + 1A + 1M + 1A + 6M + 1A + 3M + 1A

**Middle-Layer Incremental Search**

- Small scale (1.1M parameters): 3A + 1M + 1A + 3M + 2A + 1M + 1A + 3M + 1A + 2M + 3A + 2M + 1A + 1M + 1A + 6M
- Large scale (1B parameters): 3A + 1M + 1A + 3M + 2A + 1M + 1A + 3M + 1A + 2M + 3A + 2M + 1A + 1M + 1A + 6M

### B.2    Datasets Exploration: Hybrid LLM Architectures

In § 3.1, we explore the efficacy of three datasets for small-scale search: MAD (Poli et al., 2024), Sampled-DCLM (Li et al., 2024), and BabiStories (Zhang et al., 2025). We describe the hybrid LLM architecture produced by each dataset before and after extrapolation to 1B scale.

**MAD Stacking**

- Small scale (400K parameters): 2A + 4M
- Large scale (1B parameters): 4×(2A+4M) + 1A + 2M

**MAD Stretching**

- Small scale (1.1M parameters): 2A + 5M + 2A + 3M + 1A + 3M
- Large scale (1B parameters): 4A + 9M + 4A + 5M + 2A + 5M

**Small-Scale DCLM**

- Small scale (2.9M parameters): 1M + 1A + 2M
- Large scale (1B parameters): 6×(1M + 1A + 2M) + 1M + 1A + 1M

**Large-Scale DCLM**

- Small scale (500M parameters): 1A + 2M + 3A + 3M + 1A + 1M + 2A + 3M
- Large scale (1B parameters): 2A + 4M + 6A + 6M + 2A + 2M + 4A + 6M

**Small-Scale BabiStories**

- Small scale (1M parameters): 14M + 1A + 1M
- Large scale (1B parameters): 19M + 2A + 2M

**Large-Scale BabiStories**

- Small scale (150M parameters): 4A + 1M + 4A + 1M + 3A + 3M
- Large scale (1B parameters): 11A + 3M + 11A + 3M + 9A + 9M

### B.3 AGGREGATION EXPLORATION: HYBRID LLM ARCHITECTURES

In Appendix D.5, we explore the efficacy of different aggregation techniques with $N_c$ clustering. Here, we detail the hybrid LLM architecture produced by each technique for a 6-layer (400K parameters) and 16-layer search (1M parameters).

**6-Layer Search**

- $N_0$ architecture: 2A + 4M
- $N_1$ architecture: 2A + 1M + 1A + 2M
- $N_{i-1}$ architecture: 2A + 1M + 3A
- p100 architecture: 4A + 2M

**16-Layer Search**

- $N_0$ architecture: 2A + 5M + 2A + 3M + 1A + 3M
- $N_1$ architecture: 2A + 5M + 2A + 6M + 1A
- $N_{i-1}$ architecture: 2A + 4M + 3A + 2M + 4A + 1M
- p100 architecture: 3A + 7M + 1A + 5M

### B.4 EXTRAPOLATION METHODOLOGY: HYBRID LLM ARCHITECTURES

In § 3.3, we study the efficacy of stretching and stacking as we ablate $n$, the number of layers for search. We also study the efficacy of with and without width scaling. We describe the Composite architectures produced by each methodology before and after extrapolation to 1B scale below.

$n = 4$ **Layer Search**

- Small scale (270K parameters): 2A + 2M
- Large scale via stacking (1B parameters): 8×(2A + 2M) + 1A + 1M
- Large scale via stretching (1B parameters): 18A + 18M

$n = 5$ **Layer Search**

- Small scale (330K parameters): 2A + 3M
- Large scale via stacking (1B parameters): 6×(2A + 3M)
- Large scale via stretching (1B parameters): 12A + 18M

$n = 6$ **Layer Search**

- Small scale (400K parameters): 2A + 4M

- Large scale via stacking (1B parameters): $4\times(2A+4M) + 1A + 2M$
- Large scale via stretching (1B parameters): $10A+20M$

$n = 7$ **Layer Search**

- Small scale (465K parameters): $2A + 5M$
- Large scale via stacking (1B parameters): $3\times(2A + 5M) + 2A + 4M$
- Large scale via stretching (1B parameters): $8A + 20M$

$n = 8$ **Layer Search**

- Small scale (530K parameters): $2A + 4M + 1A + 1M$
- Large scale via stacking (1B parameters): $3\times(2A + 4M + 1A + 1M) + 2A + 3M + 1A + 1M$
- Large scale via stretching (1B parameters): $8A + 16M + 4A + 4M$

$n = 9$ **Layer Search**

- Small scale (600K parameters): $2A + 4M + 1A + 2M$
- Large scale via stacking (1B parameters): $3\times(2A + 4M + 1A + 2M)$
- Large scale via stretching (1B parameters): $6A + 12M + 3A + 6M$

$n = 10$ **Layer Search**

- Small scale (660K parameters): $2A + 4M + 1A + 3M$
- Large scale via stacking (1B parameters): $2\times(2A + 4M + 1A + 3M) + 2A + 3M + 1A + 2M$
- Large scale via stretching (1B parameters): $6A + 12M + 3A + 9M$

$n = 11$ **Layer Search**

- Small scale (730K parameters): $2A + 4M + 1A + 2M + 1A + 1M$
- Large scale via stacking (1B parameters): $2\times(2A + 4M + 1A + 2M + 1A + 1M) + 2A + 3M + 1A + 2M + 1A + 1M$
- Large scale via stretching (1B parameters): $6A + 11M + 3A + 6M + 3A + 3M$

$n = 12$ **Layer Search**

- Small scale (795K parameters): $2A + 5M + 1A + 4M$
- Large scale via stacking (1B parameters): $2\times(2A + 5M + 1A + 4M) + 1A + 1M + 1A + 1M$
- Large scale via stretching (1B parameters): $4A + 11M + 2A + 8M$

$n = 13$ **Layer Search**

- Small scale (860K parameters): $3A + 7M + 1A + 2M$
- Large scale via stacking (1B parameters): $2\times(3A + 7M + 1A + 2M) + 1A + 1M + 1A + 1M$
- Large scale via stretching (1B parameters): $7A + 15M + 3A + 5M$

$n = 14$ **Layer Search**

- Small scale (925K parameters): $2A + 5M + 4A + 1M + 1A + 1M$
- Large scale via stacking (1B parameters): $2\times(2A + 5M + 4A + 1M + 1A + 1M) + 1A + 2M + 2A + 1M + 1A + 1M$
- Large scale via stretching (1B parameters): $5A + 12M + 10A + 3M + 3A + 3M$

$n = 15$ **Layer Search**

- Small scale (990K parameters): $3A + 2M + 2A + 5M + 1A + 2M$
- Large scale via stacking (1B parameters): $2\times(3A + 2M + 2A + 5M + 1A + 2M)$
- Large scale via stretching (1B parameters): $6A + 4M + 4A + 10M + 2A + 4M$

$n = 16$ **Layer Search**

- Small scale (1.1M parameters): $2A + 5M + 2A + 3M + 1A + 3M$

- Large scale via stacking (1B parameters): 2A + 5M + 2A + 3M + 1A + 3M + 2A + 4M + 2A + 2M + 1A + 2M
- Large scale via stretching (1B parameters): 4A + 9M + 4A + 5M + 2A + 5M

$n = 20$ **Layer Search**

- Small scale (1.3M parameters): 2A + 9M + 1A + 1M + 1A + 1M + 1A + 2M + 2A
- Large scale via stacking (1B parameters): 2A + 9M + 1A + 1M + 1A + 1M + 1A + 2M + 3A + 4M + 1A + 1M + 1A + 1M + 1A + 1M + 1A
- Large scale via stretching (1B parameters): 3A + 13M + 2A + 2M + 2A + 2M + 2A + 3M + 3A

$n = 24$ **Layer Search**

- Small scale (1.6M parameters): 2A + 1M + 2A + 1M + 1A + 1M + 1A + 2M + 3A + 1M + 6A + 3M
- Large scale via stacking (1B parameters): 2A + 1M + 2A + 1M + 1A + 1M + 1A + 2M + 3A + 1M + 6A + 3M + 2A + 1M + 2A + 1M + 1A + 1M + 1A + 2M + 2A + 1M + 4A + 2M
- Large scale via stretching (1B parameters): 4A + 2M + 4A + 2M + 2A + 2M + 2A + 4M + 6A + 2M + 12A + 6M

$n = 28$ **Layer Search**

- Small scale (1.9M parameters): 2A + 3M + 1A + 2M + 3A + 4M + 3A + 1M + 6A + 2M + 1A
- Large scale via stacking (1B parameters): 2A + 3M + 1A + 2M + 3A + 4M + 3A + 1M + 6A + 2M + 2A + 1M + 1A + 1M + 1A + 2M + 1A + 1M + 2A + 1M + 1A
- Large scale via stretching (1B parameters): 3A + 4M + 2A + 3M + 4A + 6M + 4A + 2M + 8A + 3M + 2A

$n = 32$ **Layer Search**

- Small scale (2.1M parameters): 2A + 3M + 1A + 1M + 1A + 1M + 2A + 5M + 1A + 3M + 1A + 1M + 2A + 1M + 1A + 2M + 1A + 1M + 2A
- Large scale via stacking (1B parameters): 2A + 3M + 1A + 1M + 1A + 1M + 2A + 5M + 1A + 3M + 1A + 1M + 2A + 1M + 1A + 2M + 1A + 1M + 2A
- Large scale via stretching (1B parameters): 2A + 3M + 1A + 1M + 1A + 1M + 2A + 5M + 1A + 3M + 1A + 1M + 2A + 1M + 1A + 2M + 1A + 1M + 2A

**Width Scaling**

- Small scale (400K parameters): 2A + 4M
- Large scale (1B parameters): 4×(2A+4M) + 1A + 2M

**No Width Scaling**

- Small scale (200M parameters): 3A + 3M
- Large scale (1B parameters): 5×(3A+3M) + 2A + 2M

## B.5 COMPOSER'S BEST HYBRID LLM ARCHITECTURES ACROSS MODEL SIZES

With finalized methodologies from our design exploration, we conduct a 6-layer and 16-layer search and discover two unique architectures, presented in Equation 4 and Equation 5. We extrapolate these architectures to various sizes via stacking and stretching, respectively. Below, we describe each architecture per size.

**Stacked Composite Architecture**

- Small-scale 6-layer search: 2A + 4M
- **350M**: 4×(2A + 4M)
- **1B**: 4×(2A+4M) + 1A + 2M
- **3B**: 8×(2A+4M) + 1A + 2M
- **8B**: 10×(2A + 4M) + 1A + 1M

**Stretched Composite Architecture**

- Small-scale 16-layer search: 2A + 5M + 2A + 3M + 1A + 3M
- **350M:** 3A + 8M + 3A + 5M + 2A + 5M
- **1B:** 4A + 9M + 4A + 5M + 2A + 5M
- **3B:** 7A + 16M + 7A + 10M + 4A + 10M
- **8B:** 8A + 19M + 8A + 12M + 4A + 12M

### B.6 BASELINES

We compare our two best architectures against several baselines: Llama 3.2, Sandwich Transformer (Press et al., 2020), and Striped Attention (Poli et al., 2024). We describe each of their architectures below.

**Llama 3.2**

- **350M:** 14×(1A + 1M)
- **1B:** 16×(1A + 1M)
- **3B:** 28×(1A + 1M)
- **8B:** 36×(1A + 1M)

**Striped Attention (1B)**

- 1:2 ratio: 9×(1A + 2M)
- 1:4 ratio: 5×(1A + 4M)
- 1:8 ratio: 2×(1A + 8M) + 1A + 4M

**Sandwich Transformer (1B)**: 8A + 1M + 1A + 1M + 1A + 1M + 1A + 1M + 1A + 1M + 1A + 1M + 1A + 1M + 1A + 1M + 1A + 8M

### B.7 RANDOMLY GENERATED MODELS

In § 4.4, we assess Composer's robustness by comparing the model quality of our Composite LLMs with five randomly generated LLMs. We detail the architecture of each randomly generated LLM at 1B scale below.

- Random 1: 3M + 2A + 7M + 2A + 10M
- Random 2: 2A + 2M + 4A + 4M + 7A + 4M + 3A + 4M + 3A + 3M
- Random 3: 7A + 2M + 2A + 2M + 2A + 2M + 5A + 9M + 2A + 2M
- Random 4: 5M + 9A + 2M + 4A + 2M + 2A + 9M + 2A
- Random 5: 5A + 3M + 3A + 7M + 7A + 5M + 3A + 3M + 5A

### B.8 RELATIVE RANKINGS AT SMALL-SCALE SEARCH AND LARGE-SCALE PRE-TRAINING

In § 4.4 and Appendix D.2, we asses Composer's robustness by comparing the relative rankings of candidate architectures evaluated during small-scale search and large-scale pre-training. We detail the architecture of each randomly generated LLM before and after scaling to 1B size.

**p0 6-Layer Search**

- Small-scale (400K parameters): 6M
- Large-scale (1B parameters): 20M

**p25 6-Layer Search**

- Small-scale (400K parameters): 6A
- Large-scale (1B parameters): 96A

**p50 6-Layer Search**

- Small-scale (400K parameters):3M + 1A + 1M + 1A
- Large-scale (1B parameters): 4×(3M + 1A + 1M + 1A) +1M + 1A + 1M + 1A

**p75 6-Layer Search**

| Training Parameter | MAD | Sampled-DCLM | BabiStories |
|---|---|---|---|
| Optimizer | AdamW | AdamW | AdamW |
| Optimizer Momentum | $\beta_1, \beta_2 = 0.9, 0.98$ | $\beta_1, \beta_2 = 0.9, 0.98$ | $\beta_1, \beta_2 = 0.9, 0.98$ |
| Dropout | None | 0.05 | None |
| Batch Size | 128 | 1 | 128 |
| Vocab Size | 16-64 | 128K | 50K |
| Training Epochs | 200 | 1 | 1 |
| Learning Rate Schedule | Cosine Decay | Cosine Decay | Cosine Decay |
| Number of Training Samples | 800 | 10000 | 927158 |
| Number of Evaluation Samples | 1280 | 9275 | 9275 |
| Parallelism | None | FSDP 8 GPUs | None |
| Base Learning Rate | [1e-4, 5e-4, 1e-3] | 5e-3 | [1e-4, 5e-4, 1e-3] |
| Weight Decay | [0.0, 0.1] | 0.1 | [0.0, 0.1] |

Table 4: Composer Evaluator's training setup per dataset.

- Small-scale (400K parameters): 1M + 2A + 1M + 2A
- Large-scale (1B parameters): 7×(1M + 2A + 1M + 2A) + 1M + 1A

**p100 6-Layer Search**

- Small-scale (400K parameters): 4A + 2M
- Large-scale (1B parameters): 7×(4A + 2M) + 1A + 1M

**p0 16-Layer Search**

- Small-scale (1M parameters): 2M + 4A + 3M + 4A + 1M + 1A + 1M
- Large-scale (1B parameters): 5M + 9A + 7M + 9A + 3M + 3A + 3M

**p25 16-Layer Search**

- Small-scale (1M parameters): 1M + 2A + 2M + 5A + 1M + 2A + 3M
- Large-scale (1B parameters): 3M + 5A + 5M + 12A + 3M + 5A + 7M

**p50 16-Layer Search**

- Small-scale (1M parameters): 2A + 1M + 1A + 5M + 2A + 1M + 3A + 1M
- Large-scale (1B parameters): 5A + 3M + 3A + 10M + 5A + 3M + 6A + 3M

**p75 16-Layer Search**

- Small-scale (400K parameters):2A + 3M + 4A + 2M + 1A + 2M + 1A + 1M
- Large-scale (1M parameters): 5A + 6M + 8A + 5M + 3A + 5M + 3A + 3M

**p100 16-Layer Search**

- Small-scale (1M parameters): 3A + 7M + 1A + 5M
- Large-scale (1B parameters): 5A + 11M + 2A + 8M

## C  EXPERIMENTAL DETAILS

### C.1  SEARCH TRAINING/VALIDATION SETUP

During small-scale search, the HNAS Evaluator evaluates each candidate hybrid LLM architecture provided by the HNAS Engine. We provide details on the training and validation setup the HNAS evaluator uses. This setup changes depending on the dataset used for evaluation. Table 4 details the training setup for each dataset. We explore three datasets: MAD (Poli et al., 2024), Sampled-DCLM (Li et al., 2024), and BabiStories (Zhang et al., 2025). For MAD and BabiStories, we follow the same training setup (Poli et al., 2024). For Sampled-DCLM, we reduce the sample size 12K samples.

| Model Size | DP | Local Batch Size | Acc | Seq Len | Global Batch Size | Learning Rate |
|---|---|---|---|---|---|---|
| 350M | 8 | 8 | 1 | 8192 | 0.5B | 3e-3 |
| 1B | 16 | 4 | 1 | 8192 | 0.5B | 6e-4 |
| 3B | 16 | 4 | 1 | 8192 | 0.5B | 3e-4 |
| 8B | 16 | 4 | 1 | 8192 | 0.5B | 6e-4 |

Table 5: Pre-training setup details per model size.

## C.2 MODEL SIZE AND FLOPS CALCULATION

We describe size (number of parameters) and FLOPs calculations per computational primitive: Grouped Query Attention (GQA) (Ainslie et al., 2023) and SwiGLU (Shazeer, 2020).

**Attention:** We provide a general formulation for calculating FLOPs (forward and backward pass) and parameter size of a single Attention layer that can be applied to both GQA or MHA layers. We follow DeepMind's calculations (Hoffmann et al., 2022), calculating FLOPs for the forward and backward pass.

$$\text{Attn Parameter Count} = 2 \times dim^2 + 2 \times \frac{dim}{num\_heads} \times num\_kv\_heads \tag{6}$$

$$\begin{aligned}\text{Attn FLOPs} = 6 \times dim^2 + 6 \times 2 \times \frac{dim}{num\_heads} \times num\_kv\_heads \\ + 6 \times dim^2 + 3 \times 2 \times 2 \times num\_heads \times dim\end{aligned} \tag{7}$$

**MLP (SwiGLU):**
$$\text{MLP Parameter Count} = 3 \times dim \times hidden\_dim \tag{8}$$

$$\text{MLP FLOPs} = 6 \times 3 \times dim \times hidden\_dim \tag{9}$$

## C.3 SCALING LAWS PRE-TRAINING SETUP

For fair comparison between models, we keep training setups between architecture as similar as possible per model size. We pre-train all models using TorchTitan (Liang et al., 2025), adding support for custom LLM architectures. Table 5 details the pre-training setup. To study how our models scale, we compare across four model sizes: 350M, 1B, 3B, and 8B models. We use FSDP parallelism (Zhao et al., 2023) to train all models. Models of sized 350M are trained on 8 H200 GPUs, while remaining sizes are trained on 16 H200 GPUs. We linearly interpolate learning rates from common settings, obtaining a linear inverse relationship with model size. For all models, we train with the DCLM dataset (Li et al., 2024) with 0.5B global batch size and sequence length 8192. We use the AdamW optimizer. Following (Bae et al., 2025), we use a Trapezoidal learning rate scheduler (Xing et al., 2018), with linear warm up and decay phases using 20% the number of training steps each.

To determine the length of training, we use the IsoFLOP methodology, like DeepMind's Chincilla (Hoffmann et al., 2022), and fix the training budget (number of FLOPs to train with) across hybrid architectures: five budgets of 2e19, 4e19, 8e19, 2e20, 4e20 FLOPs. As the training budget and model sizes between architectures are the same, the only differences in training between the models is the number of training tokens. This is because the FLOPs per token differ between models, since the ratio of computational primitives can vary. Appendix C.2 details equations for calculating the FLOPs of a single Attention or MLP layer in the forward and backward pass processing one token. The FLOPs per token of a given architecture is then given by:

$$\begin{aligned}\text{FLOPs per token } (M) = (\text{Num Attn layers}) \times (\text{Attn FLOPs}) \\ + (\text{Num MLP layers}) \times (\text{MLP FLOPs})\end{aligned} \tag{10}$$

The number of training tokens $D$ for a given model $M$ under a given training budget $C$ FLOPs is

$$D = \frac{C}{\text{FLOPs per token } (M)} \tag{11}$$

### C.4 INFERENCE EFFICIENCY MEASUREMENT PROTOCOL

To evaluate Composer's improvements in inference efficiency with its discovered hybrid LLMs, we measure the per-token generation time. We measure the generation time under various model configurations using random weights and inputs. We evaluate throughput and inference latency with a variety of sequence lengths—1K, 2K, 4K, 8K, 16K, 32K—where 25% of the sequence length is the prompt. We use three prompts as a warm up phase, and then measure the average throughput (tokens per second) and latency across an additional five prompts.

### C.5 COMPARISON AGAINST PREVIOUS SOTA WORKS PRE-TRAINING SETUP

In § 4.3, we follow the pre-training setup of STAR's (Thomas et al., 2025) 1B experiment results to fairly compare against them. Rather than fixing the training FLOP budget, we pre-train every model with the same number of tokens instead: 37.5B tokens (see Table 5 of 1B setup). All models are pre-trained for 71564 training steps.

### C.6 DOWNSTREAM TASKS

We evaluate validation loss on the test set from the DCLM dataset (Li et al., 2024). Additionally, we use the Language Model Evaluation Harness framework (Gao et al., 2024) to evaluate accuracy on six few-shot tasks: HellaSwag (Zellers et al., 2019), PIQA (Bisk et al., 2020), WinoGrande (Sakaguchi et al., 2021), ARC-easy and ARC-challenge (Clark et al., 2018), and SciQ (Welbl et al., 2017). We adhere to the standard number of shots specified by the evaluation framework for each task. All evaluate performance measurements were conducted on a single H200 GPU.

## D ADDITIONAL RESULTS

### D.1 SCALING LAWS ANALYSIS

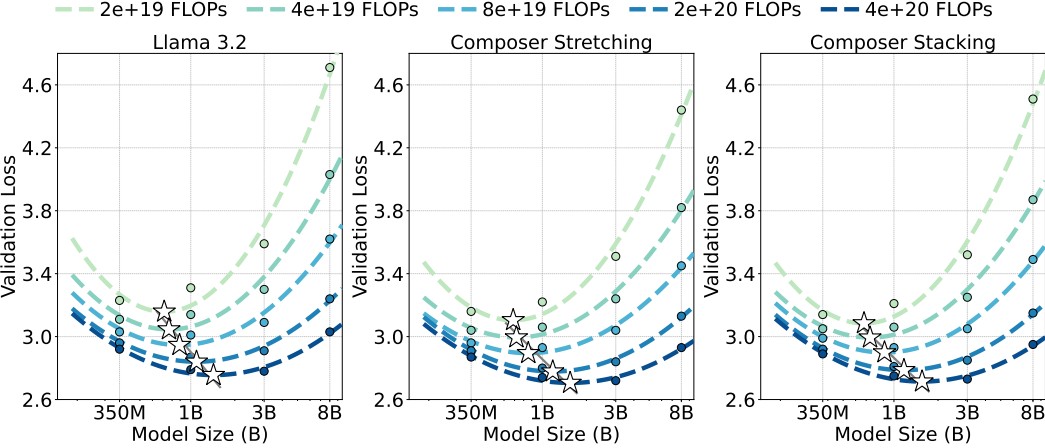

Figure 8: Compute-optimal scaling analysis for Llama 3.2 and our Composite LLMs. Each star indicates the optimal model size for a given compute budget.

Figure 8 and 9 illustrates that our Composite LLMs exhibit a very similar compute-optimal scaling behavior compared to Llama 3.2 under IsoFLOPs constraints. In Figure 9, the Composite models have nearly identical slopes (Llama has a slope of -0.63, where Stretching and Stacking have slopes of -0.628 and -0.612, respectively), indicating stable gains in performance as we scale model sizes. While further investigation is required to validate that Composer's extrapolation techniques scale to larger model sizes (e.g., tens to hundreds of billions of parameters), our work lays a blueprint for efficiently discovering new hybrid LLMs that can scale $1000\times$ the search size while still outperforming Transformer-based architectures.

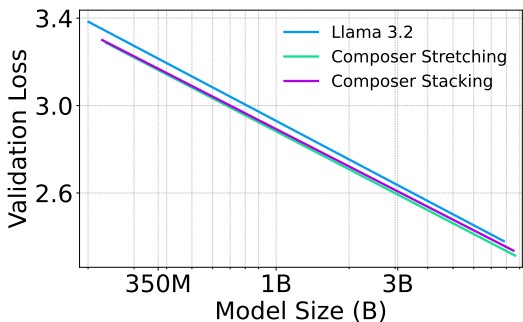

Figure 9: Compute-optimal scaling analysis for Llama 3.2 and our Composite LLMs.

## D.2 COMPOSER'S ROBUSTNESS: RELATIVE RANKINGS

In § 4.4, we show Composer's search framework is robust, as the relative rankings of explored 6-layer candidate architectures during small-scale during search and at 1B scale after pre-training are nearly identical. We provide additional results detailing the relative rankings for 16-layer search in Figure 10. The Spearman rank correlation between small-scale 16-layer search is 0.9 on average across compute budgets, indicating high positive correlation (nearly identical ranking). Appendix B.8 details each architecture. This high correlation in relative-rankings between small-scale search and at-scale performance showcases our search framework's robustness.

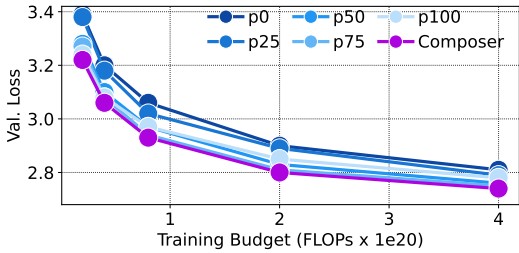

Figure 10: Performance of ranked candidate architectures during 16-layer search (p0-p100) at 1B scale stretched out. Relative rankings between small and large scale hold.

## D.3 EXTENDED INFERENCE EFFICIENCY RESULTS

In § 4.2, we detail the inference efficiency improvements of our Composite LLM's compared to Llama 3.2 at 1B scale with a batch size of 1. We provide further analysis across other model sizes 350M to 3B with batches of 1, 4, 8, and 16 prompts. Figure 11- 13 present the KV cache size (top), inference throughput (middle), and inference latency (bottom) as sequence length varies. Across model sizes and batch sizes, our Composite LLMs' continue to reduce KV cache size, increase infernece throughput, and reduce inference latency as sequence length increases.

## D.4 PERFORMANCE ON DOWNSTREAM TASKS ACROSS MODEL SIZES

In § 4.1, we report the DCLM validation loss our Composite LLMs against Llama 3.2 across four sizes: 350M, 1B, 3B, and 8B. We provide additional results in Table 6 that shows that across model sizes, our Composite LLMs consistently improve accuracy on 6 downstream tasks (tasks are detailed in Appendix C.6). We pre-train each model with the max training budget (4e20 FLOPs) using the DCLM (Li et al., 2024) dataset. We outperform Llama 3.2 in all downstream tasks with performance improvements up to 2.8-8.3% (1.1-3.1% avg).

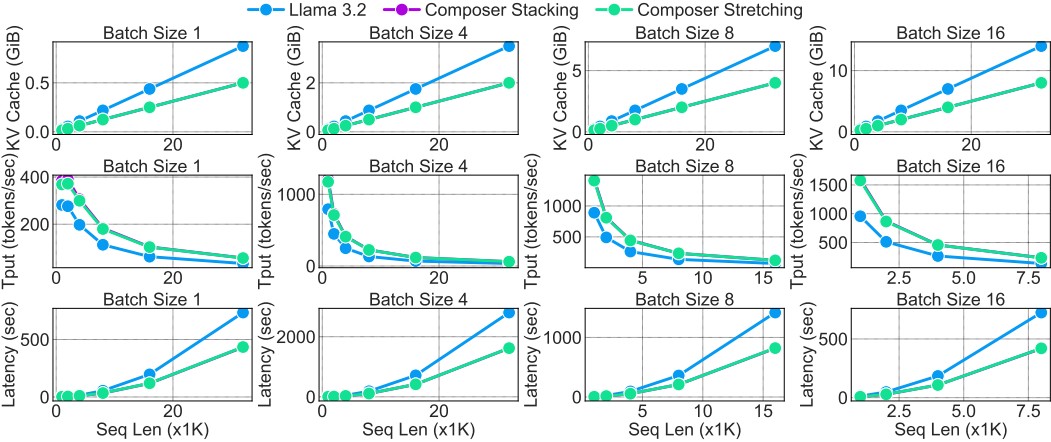

Figure 11: Comparison of the inference efficiency of our Composite LLMs versus Llama 3.2 at 350M scale. We report KV cache size (top), inference throughput (middle), and inference latency (bottom) as prompt lengths and batch size change.

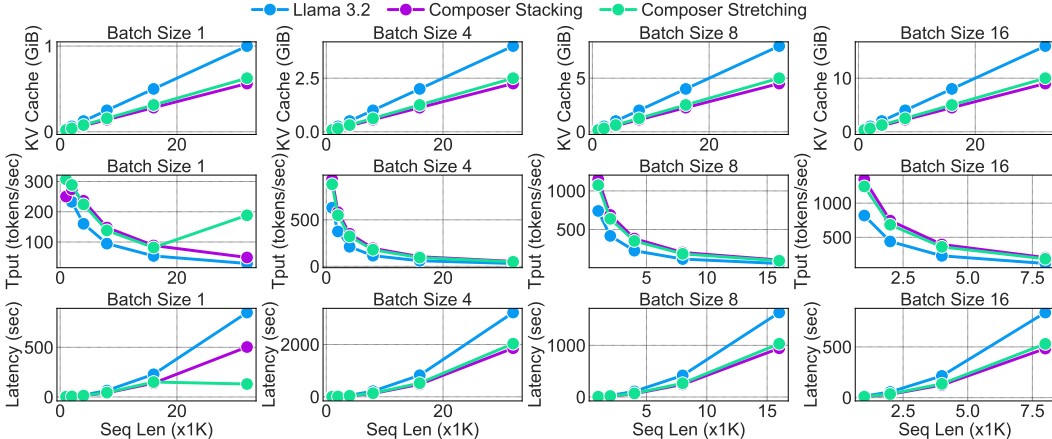

Figure 12: Comparison of the inference efficiency of our Composite LLMs versus Llama 3.2 at 1B scale. We report KV cache size (top), inference throughput (middle), and inference latency (bottom) as prompt lengths and batch size change.

## D.5 EXPLORATION OF AGGREGATION TECHNIQUES

After the HNAS Engine and Evaluator finish searching and evaluating the design space of LLM architectures, the HNAS Aggregator leverages $N_c$ clustering to synthesize the final hybrid LLM architecture from the search results. § 2.3 explains the algorithm for $N_c$ clustering.

In this section, we evaluate three different values for $c$. When $c = 0$, $N_0$ clustering selects the dominant block at each layer among the top candidate architectures independently, resulting in no conditioning based on prior layers. When $c = 1$, $N_1$ clustering conditions the block choice at each layer on the block selected at the immediate preceding layer. Finally, for $c = i - 1$, the primitive selected at each layer index $i$ is conditioned on the entire sequence of previously selected blocks, enforcing full prefix consistency.

To convincingly understand which aggregation technique is best, we exhaustively evaluate all hybrid $n$-layer architectures at small scale using MAD (Poli et al., 2024), from $n = 4$ to 10. Then, we apply $N_0$, $N_1$, and $N_{i-1}$ for each value of $n$ to synthesize the final hybrid architecture and scale up to 1B scale via stacking. We also scale up the p100 (best) model during small-scale evaluation. Finally, we include results for 16-layer search, however, we do not exhaustively evaluate all $2^{16}$ hybrid

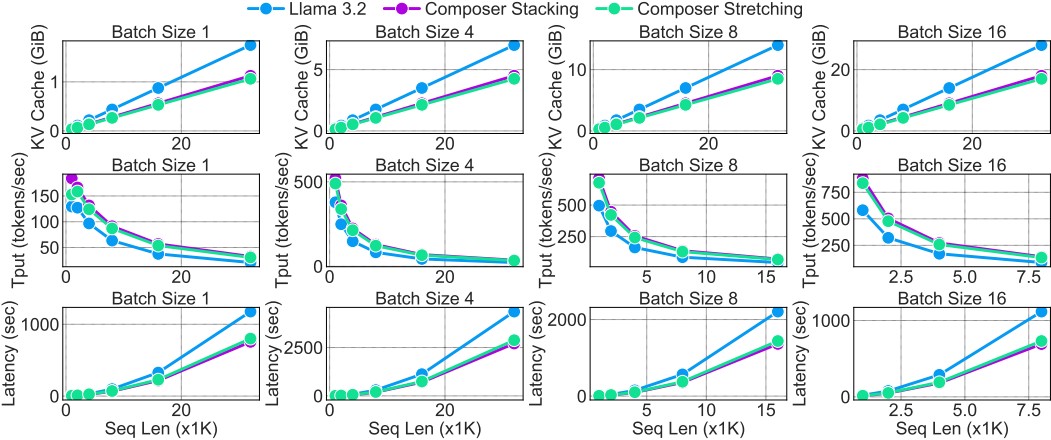

Figure 13: Comparison of the inference efficiency of our Composite LLMs versus Llama 3.2 at 3B scale. We report KV cache size (top), inference throughput (middle), and inference latency (bottom) as prompt lengths and batch size change.

| Size | Model | Arc C. | Arc E. | Hella. | Wino. | SciQ | PIQA | Avg. |
|------|-------|--------|--------|--------|-------|------|------|------|
| 350M | Llama 3.2 | 27.7 | 53.2 | 49.2 | 54.1 | 78.3 | 70.1 | 55.4 |
| | Composite - Stacked | 27.9 | 53.9 | 49.6 | **54.8** | 78.7 | **71** | 56.0 |
| | Composite - Stretched | **30.5** | **55.4** | **50** | 54.4 | **79.6** | 70.3 | **56.7** |
| 1B | Llama 3.2 | 30 | 56.8 | 52.9 | 55.9 | 81.7 | 70.7 | 58.0 |
| | Composite - Stacked | **33.5** | 59.5 | **56.7** | **59.4** | 83.2 | **73.1** | 60.9 |
| | Composite - Stretched | 32.5 | **61.5** | 56.1 | 56.1 | **88.1** | 72.5 | **61.1** |
| 3B | Llama 3.2 | 30.1 | 58.6 | 54.3 | 56.4 | 83.1 | 72.8 | 59.2 |
| | Composite - Stacked | 31.5 | 60.3 | 57.1 | 57.2 | **85.2** | **74.4** | 61.0 |
| | Composite - Stretched | **33.5** | **66.9** | **57.9** | **57.5** | 84 | 72.3 | **62.0** |
| 8B | Llama 3.2 | 42.2 | 70.9 | 70.2 | 66.0 | 90.5 | 76.5 | 69.3 |
| | Composite - Stacked | 42.7 | 71.5 | **70.6** | 64.8 | 91 | 77.4 | 69.7 |
| | Composite - Stretched | **43.6** | **71.8** | 70.4 | **66.5** | **91.3** | **78.5** | **70.4** |

Table 6: Evaluation of Composer's Stacked and Stretched hybrid architectures compared to Llama 3.2 on 6 downstream tasks.

architectures during small scale, as it is prohibitive. Instead, results presented are aggregated from 100 search trials. Figure 14 details the DCLM validation loss per hybrid LLM produced by each aggregation technique across $n = 4$ to $n = 10$-layer search.

Across all $n$-layer search, $N_0$ clustering synthesizes the highest quality hybrid LLM. Increasing $c$ reduces the number of top candidate architectures. $N_c$ clustering aggregates over to synthesize a hybrid LLM, thereby reducing the generalizability of the produced LLMs to at-scale performance. $N_0$ clustering also outperforms the p100 model, as it smooths over any noise or overfitting phenomena that occurs during search. Hence, Composer leverages $N_0$ clustering to produces its two Composite LLMs.

The following explanation provides further intuitive reasoning as to why $N_0$ clustering works well. Consider each architecture as a sequence of blocks across layer indices. The search process returns a set of top-performing architectures, which we filter based on validation accuracy with MAD. We interpret this collection as a representative Monte Carlo sample from a distribution over high-performing designs. This distribution is implicitly shaped by the architecture search process — in our case, Bayesian optimization (BO). The BO surrogate model evaluates full architectural configurations and captures interactions between blocks at different layers, learning correlations between structural choices and validation performance. As a result, the search does not sample randomly, but instead includes architectures that exhibit useful inter-layer dependencies.

Given this, we can define an empirical frequency for how often each block appears at a given layer across the sampled architectures. This frequency estimates how likely that block is to appear at that

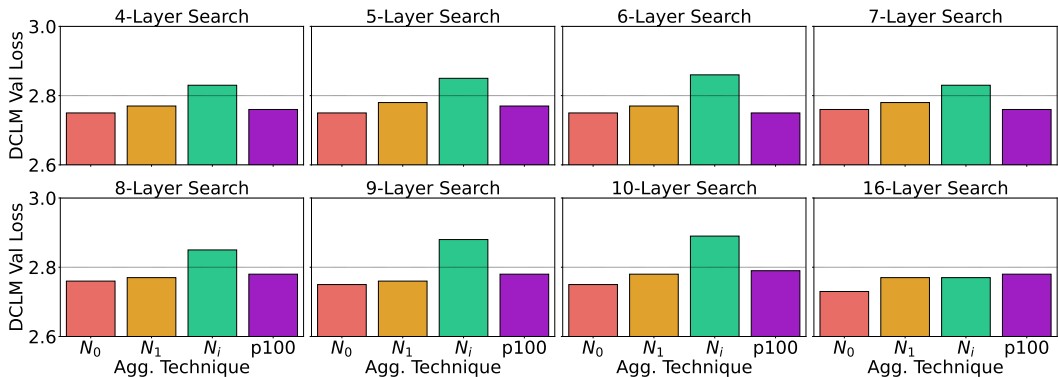

Figure 14: Comparison of $N_c$ aggregation techniques across different $n$-layer searches.

layer in a high-performing design. $N_0$ clustering selects, for each layer, the block that appears most frequently. The final architecture is then assembled by independently choosing the most common block at each layer across the top-performing candidates.

Selecting the block with the highest marginal frequency at each layer is equivalent to computing the architecture that maximizes the product of these marginal frequencies across all layers. In other words, $N_0$ clustering selects a configuration that is most probable under the assumption that each layer's block choice is independent of the others. This corresponds to a Naive Bayes-style estimator — a well-known technique in probabilistic modeling — which maximizes the product of marginal likelihoods for individual variables while ignoring conditionals.

While this formulation ignores explicit inter-layer dependencies during aggregation, we argue this is appropriate because such dependencies are already accounted for during the search phase. That is, the BO search process acts as a dependency-aware sampler to construct the set of top-performing architectures. $N_0$ clustering aggregates across a structurally pre-filtered space, smoothing out noisy or overfitting samples while preserving dominant patterns.

Under this framing, $N_0$ clustering is a statistically consistent estimator of the most likely block sequence under a marginal model of high-performing architectures. By prioritizing frequent blocks and avoiding conditional noise or overfitting, $N_0$ clustering yields architectures that generalize well when extrapolated to larger scales.

### D.6    EXPLORATION OF DATASETS: BABISTORIES

In our main paper, we explored the efficacy of using two different datasets (MAD (Poli et al., 2024) and a sampled-down version of DCLM (Li et al., 2024)) for evaluating candidate hybrid architecture with the HNAS Evaluator. We also include our findings when using BabiStories (Zhang et al., 2025) to evaluate candidate architectures. As we leverage MAD (Poli et al., 2024) throughout our paper, we include MAD's results here as well for comparison.

Similar to our experiments with DCLM in § 3.2, we explored two approaches for leveraging BabiS-tories during small-scale search. First, we randomly sample down the BabiStories dataset while keeping model size large (150M parameters). We perform a 16-layer search and extrapolate the discovered hybrid LLM to 1B parameters via stretching. Figure 15 shows that this methodology, *Large-Scale Stories*, yields a relatively high quality model that outperform Llama 3.2 across training budgets. However, the search cost is large (>100 GPU-hours). Hence, as a second approach, we reduce the width of the model and keep model size between 1M-2M parameters, *Small-Scale BabiS-tories*. However, the stretched model at 1B scale consistently performs worse than Llama 3.2 across all training budgets. Ultimately, like DCLM, we found that using BabiStories was either impractical or produced poor-performing models. Thus, Composer's HNAS Evaluator uses MAD throughout this work to quickly but accurately evaluate the efficacy of candidate LLM architectures.

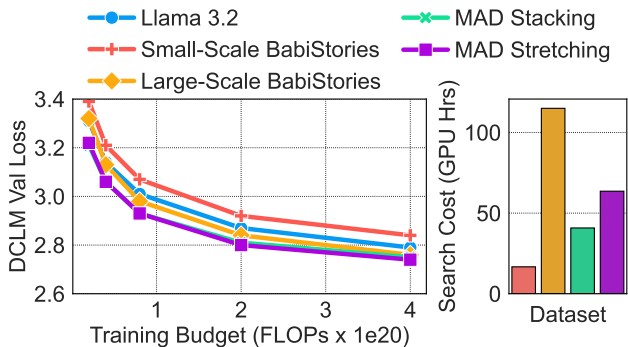

Figure 15: Design exploration of datasets for HNAS Evaluator. We report the model quality at 1B scale and search cost for each technique.

## E   COMPOSER DESIGN DETAILS

### E.1   VISUALIZATION OF BAYESIAN OPTIMIZATION SEARCH JOURNEY

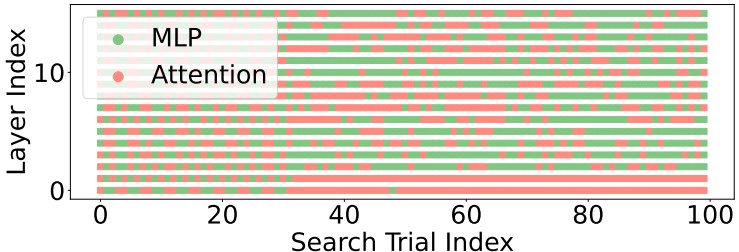

Figure 16: Pictoral representation of Composer' Search Engine converging on the computational primitives per layer index over 100 search trials for a 16-layer search.

Figure 16 shows Composer's Search Engine converging on computational primitives per layer index over 100 search trials for a 16-layer search. It discovers to begin the hybrid architecture with attention layers and end with MLP layers. It then explores different interleavings of layers in between to find top performing hybrid architecture candidates.

### E.2   DESCRIPTION OF DATASETS USED BY THE HNAS EVALUATOR

**MAD Dataset (Poli et al., 2024):** The MAD dataset is a set of synthetic tasks. The dataset consists of simple pretext token manipulations that provide quick and cheap performance estimates for a given LLM arhcitecture. MAD consists of six categories of tasks that LLMs should be capable of performing: in-context recall, fuzzy in-context recall, noise in-context recall, selective copying, compression, and memorization. For each candidate hybrid architecture in our search, we train the architecture across 800 samples per task and then evaluate the architecture on 1280 samples to obtain cross-entropy loss for each task. We average the cross-entropy loss across all six tasks as the final loss for the given architecture.

**BabiStories (Zhang et al., 2025):** The BabiStories dataset is a synthetic benchmark of children's stories, meant to train and evaluate the reasoning and generalization capabilities of language models in narrative contexts. The dataset is generated by LLMs using a set of rules that define entities, actions, and events, allowing for the systematic creation of diverse storylines. The simple grammar and smaller vocabulary size of the dataset allows small models of millions parameter scale to produce coherent English.

**DCLM (Li et al., 2024):** The DCLM dataset is a large-scale, curated corpus designed for training and evaluating language models at scale. It comprises over 1 million documents spanning diverse

domains, including news articles, scientific papers, and web content, each annotated with rich metadata (e.g., publication date, source, topical categories). Unlike traditional sentence- or paragraph-level datasets, DCLM preserves the full document structure, enabling models to capture long-range dependencies and discourse-level phenomena. The dataset is preprocessed to ensure high textual quality, with non-linguistic artifacts and duplicates removed.

