# OpenReview forum: "Composer: A Search Framework for Hybrid Neural Architecture Design"
_ICLR.cc/2026/Conference — ICLR 2026 Poster_

### Official Review · Reviewer_PWjK · 2025-10-30

**Soundness:** 2
**Presentation:** 2
**Contribution:** 2
**Rating:** 4
**Confidence:** 3

**Summary:**

The paper addresses the limitations of traditional Transformer architectures in large language models (LLMs), which rely on a fixed 1:1 interleaving of self-attention and multi-layer perceptron (MLP) layers.
The authors note that hybrid architectures, using varied arrangements of computational primitives such as attention and MLPs, have demonstrated superior performance in prior manual designs.
However, the vast design for a 32-layer model using only attention and MLP—coupled with high training costs, makes manual exploration intractable.
Composer introduces an automated framework for hybrid neural architecture search (HNAS) tuned to pre-training from scratch, aiming to discover architectures that outperform baselines like Llama 3.2 at scales from 350M to 3B parameters.

The framework comprises four core components:
* The Search Engine employs Bayesian Optimization with Gaussian Processes for one-shot searches or incremental methods (end-layer and middle-layer) to explore small-scale models with reduced widths.
* The Evaluator assesses these candidates using small proxy datasets to provide rapid feedback predictive of large-scale performance.
* The Aggregator generates top candidates through Nc clustering, selecting primitives layer-by-layer based on conditional frequencies among clustered architectures.
* The Extrapolator scales up the models to the target size using stacking (repeating blocks) or stretching (proportional expansion of primitive groups).

Experiments yield hybrids surpassing Llama 3.2. The authors remark the benefits of 1:2 attention-to-MLP ratios and non-standard interleavings over the standard 1:1 Transformer structure.

Proxy datasets like MAD (and other synthetic datasets like BabiStories) enable efficient evaluation, outperforming downscaled DCLM in cost.

During aggregation, Stacking is robust across search depths, but stretching excels for larger small-scale searches (e.g., 16 layers).

The resulting "Composite" architectures, consisting of 1:2 interleavings of Grouped Query Attention and SwiGLU layers, reduce validation loss by 0.05-0.08 and improve downstream task accuracy by 1.1-3.1% on average, while improving training throughput by 1.25x.

**Strengths:**

The paper introduces a principled, module framework for NAS. The framework's four components (Search Engine, Evaluator, Aggregator, and Extrapolator) provide a blueprint that is extensible to other primitives.

Through experiments, the author find "creative" 1:2 attention-to-MLP ratios and non-standard interleavings that represent useful alternative architectures for future exploration.

**Weaknesses:**

The novelty of the method is minimal, as the Composer framework consists of common building blocks.

One critical weakness of the paper is that the proposed method proposes no theoretical or empirical justification of optimality. Through experiments, the authors are able to find hybrid architectures that improve upon the LLaMA3.2 baseline, however in the absence of theoretical grounding, we would like to see the Oracle of these experiments, i.e. a large scale experiment to train all candidates to completion, in order to validate the use of proxy evaluations.

Similarly, there is no evidence that the method scales to models that are larger than 3B parameters, how do the authors justify the use of the Composer framework for these?

The paper is also limited in scope, since it only mentions in the end, but does not test the method on state-space models, or less conventional attention mechanisms.

Empirical results, while positive (e.g., 0.05-0.08 validation loss reductions and 1.1-3.1% average downstream accuracy gains), show modest and context-specific advantages. Some variants, like those from downsampled DCLM or BabiStories, only marginally outperform or even underperform Llama 3.2 at higher training budgets.

**Questions:**

Will the code be publicly available?

What is the performance of Composer-discovered hybrids on long-context tasks like needle-in-a-haystack or multi-document QA? These tasks have been shown to benefit from attention architecture improvements.

Has the framework been applied to other primitives, such as state-space models (e.g., Mamba) with attention?

---

> ### Author Response · Authors · 2025-11-20
> **Answer to Reviewer PWjK's Questions**
>
> ## Answer to Q1:
> Yes, we plan to open-source the framework code along with scripts to reproduce the results upon acceptance.
>
> ## Answer to Q2:
> We are working on fine-tuning our models for evaluating on reasoning tasks such as GSM8K and Math. We will get back with the results before the rebuttal deadline. We have few-shot evaluation results on Math, GSM8K and MMLU benchmarks without fine-tuning for your reference in Table A. While the results are low for Llama 3.2 and our Composite models (we are actively fine-tuning to improve performance), the results indicate that our Composite LLMs slightly outperform Llama 3.2 on math and reasoning tasks.
>
>
> |   Model |  Math | GSM8K |  MMLU  |
> |----------|-----:|:-----:|:------:|
> | Llama-3.2            | 2.50% | 1.67% | 31.06% |
> | Composite: Stacked   | 2.84% | 2.07% | 31.33% |
> | Composite: Stretched | 2.60% | 1.82% | 31.48% |
> Table A. Math, GSM8K, MMLU results for Llama3.2 and Composite models. Model size is 1B and compute budget is 4e20. The models are only pretrained on DCLM without post-training. MMLU score is with continuation generation.
>
> In addition to these results and fine-tuning our current Composite architectures, we note that to improve performance on the suggested disciplines, we would include other computational primitives in our search (e.g., sliding window attention), as previous works show that LLM architectures with such primitives are better suited for long-context and extended reasoning (InfLLM NeurIPS’24, CAB ICML’23) . In this work, we narrow the scope of our primitives to GQA and SwiGLU layers to first build a robust HNAS framework that can be extended for future LLM architecture design. While we have not included sliding window attention in our search, we’ve conducted experiments with other computational primitives (e.g., state space models like Mamba Mamba: Linear-Time Sequence Modeling with Selective State Spaces, COLM’24 ), and our results indicate that Composer can effectively navigate the increased search space and discover new hybrid architectures with better model quality (see answer to Q3 below).
> Moreover, the dataset we use for HNAS’s Evaluator is not designed to optimize for reasoning or long context tasks. Rather, MAD’s proxy benchmarks are designed for common language tasks. To our knowledge, no small-scale datasets exist that target reasoning/long context tasks, which ideally we would use to discover hybrid LLMs targeting such tasks. As the use of proxy datasets for quick reliable signals is a research avenue that has recently grained traction (MAD appeared in ICML’24, the TinyStories/BabiStories datasets appeared in arxiv’23/ICLR’25), Composer’s modular design enables future studies to rapidly evaluate the efficacy of such new datasets targeting other LLM tasks.
>
> ## Answer to Q3:
> Our framework is in principle agnostic to the choice of primitive block: any module with a compatible interface can be used as a candidate, including state-space models such as Mamba. We run preliminary experiments where we extend the search space to include Mamba blocks and perform a 6-block search on the MAD dataset. The top architectures selected by our method are:
> * 2 Mamba + 4 Attention
> * 3 Mamba + 3 Attention
> * 2 Mamba + 1 MLP + 3 Attention
>
> Note that all three architectures include multiple Mamba blocks when the model is free to choose any block type for every layer. This observation is consistent with the literature that shows hybrid mamba and transformer architectures outforms either of the counterparts (e.g. [Samba](https://arxiv.org/abs/2406.07522), [Mamba-2](https://arxiv.org/abs/2405.21060), [Hybrid Architectures for Language Models](https://arxiv.org/pdf/2510.04800?)). Furthermore, these hybrids differ from the hand-designed Mamba–Attention combinations explored in current literature and illustrate that our framework can automatically discover novel and non-trivial interleavings of state-space and attention layers. A systematic large-scale evaluation of these architectures (including scaling them up and tuning their training setups) is ongoing work and beyond the scope of this submission, but we will add a brief discussion of these preliminary results to the revised version to clarify that the framework naturally extends to such primitives.

---

> > ### Author Response · Authors · 2025-12-01
> > **Update: Answer to Q3 - Mamba search results**
> >
> > We have run 1B model pre-training experiments with the found models from small scale search including Mamba blocks as primitives. We follow the same setup as Table 2, using a fixed token 37.5B budget to train these models, here are the validation loss results:
> >
> > | Model | Loss |
> > |----|----|
> > | 2 Mamba + 4 Attention | 2.756 |
> > | 3 Mamba + 3 Attention | 2.755 |
> > | 2 Mamba + 1 MLP + 3 Attention | **2.743** |
> > | 1 Mamba + 1 MLP | 2.773 |
> > | Llama 3.2 | 2.776 |
> >
> > All of the hybrid models beat Llama and pure Mamba (1 Mamba + 1 MLP) baseline performance and 2 Mamba + 1 MLP + 3 Attention pattern performing the best among others. To the best of our knowledge this hybrid model pattern has not been studied in the literature before. While further scaling and evaluation results are needed, this shows further evidence that Composer is able to find unique patterns that were not explored in the literature.
> > (Note: Llama baseline is slightly different from the Table 1 of the paper due to updated TorchTitan codebase. To be consistent, the results above are done using the same updated codebase.)

---

> ### Author Response · Authors · 2025-11-20
> **Answer to Reviewer PWjK's Weaknesses**
>
> ## Answer to W1:
> Our contribution is that we are the first study to design a robust search framework in a principled manner that enables rapid discovery of new hybrid architectures. Currently, the model architecture design process is manual and based on intuition — no systematic framework exists today to enable automatic, efficient discovery of hybrid LLM architectures that perform well at scale. We are the first to systematically design a HNAS framework by studying four key design questions (outlined in Section 1). Even with the Attention and MLP building blocks alone, we discover new hybrid LLM architectures that outperform the conventional transformer architecture leveraged by SOTA LLMs, like Llama 3.2.
>
> ## Answer to W2:
> Many NAS frameworks demonstrate their efficacy via empirical analysis. Moreover, to the best of our knowledge, there is no theoretical analysis proving optimality for the stacked, 1:1 transformer architecture either [Sandwich Transformers, ACL’20]. Showing optimality empirically is extremely costly and prohibitive: for example, to prove optimality at 1B scale with DCLM requires pre-training 2^32 architecture (32 layers, 2 primitives options per layer), which is >4 billion architectures. Even proving optimality with a 16-layer architecture is infeasible, with >65K architectures. This prohibitive cost is exactly why we built Composer: to navigate the search space of hybrid architectures and efficiently discover quality LLMs.
>
> While we cannot pre-train all potential architectures at scale, Figure 6-bottom shows that the relative rankings of hybrid LLMs (labeled “p25-p100”) evaluated during small-scale search hold at large-scale when we extrapolate these architectures and pre-train them. The high Spearman correlation of 0.97 suggests our evaluation of discovering high-quality models during small-scale search is indicative of high-quality models for large-scale pre-training.
>
> ## Answer to W3:
> We collected pretraining results at 8B scale with a compute budget of 4e20 FLOPs. The results on the 6 downstream tasks shows that our Composite LLMs continue to outperform baseline Llama models. Note that the 8B architecture is produced by following the same methodology to produce the 350M-3B models.
>
> | Size | Model | Loss | Arc C.|  Arc E. |  Hella. |  Wino. |  SciQ |  PIQA | Avg. |
> |-------|--------|----|----|----|----|----|----|----|----|
> | 8B | Llama 3.2 | 3.03 | 42.2 | 70.9 | 70.2 | 66.0 | 90.5 | 76.5 | 69.3 |
> | 8B | Composite - Stacked | 2.95  | 42.7 | 71.5 | **70.6** | 64.8 | 91 | 77.4 | 69.7 |
> | 8B | Composite - Stretched | **2.93** | **43.6** | **71.8** | 70.4 | **66.5** | **91.3** | **78.5** | **70.4** |
>
> We are also exploring another methodology for extrapolating our small discovered LLMs to larger sizes (e.g., 8B). Currently, we either stretch OR stack to extrapolate to larger scales. We see that stretching outperforms stacking at 3B model size, as it maintains a quality interleaving pattern and ratio of the computational primitives. However, continuously stretching small discovered LLMs to larger sizes may degrade model quality, as large chunks of the LLM will be dominated by computational primitive (e.g., all GQA block for several layers in a row), creating a poor interleaving.
>
> There are two methodologies we propose to overcome this. We could increase the number of layers and trials in our search. However, this comes with a tradeoff of increased search cost. Hence, we propose an alternative strategy that combines both stretching and stacking. This methodology will create a high-quality 3B model via stretching that maintains a good interleaving pattern and ratio of primitives, and then, by stacking into 8B, maintain quality interleaving and ratio of primitives without large chunks of the LLM being dominated by a single primitive. We will add our findings after we pre-train the hybrid LLMs produced with this extrapolation methodology.
>
> ## Answer to W4:
> We clarify that in our final search methodology to produce our Composite LLMs, we do not use downsampled DCLM or BabiStories in the Evaluator. The purpose of including data with downsampled DCLM or BabiStories was to provide a robust design exploration of different datasets and determine which ones are most effective in providing fast, reliable signals of model quality.
> However, we also clarify that the downsampled DCLM variants do in fact, outperform Llama 3.2 across training budgets (Figure 2, right). With BabiStories, searching with 150M parameter models also produces hybrid LLMs that outperform Llama 3.2 across training budgets. However, conducting very small-scale search (1-2M param models) with this dataset performs worse than Llama 3.2. We suspect this is because the vocab size of the dataset (50K) is too large and complex for small LLMs when conducting small-scale search, and hence the small architectures are poorly trained, rendering it challenging for Composer to evaluate the efficacy of any candidate architecture.

---

### Official Review · Reviewer_naoJ · 2025-10-31

**Soundness:** 3
**Presentation:** 3
**Contribution:** 2
**Rating:** 4
**Confidence:** 4

**Summary:**

This paper presents Composer, a systematic framework for automating the design of hybrid neural architectures that combine computational primitives (e.g., Attention, MLP) in novel ratios and interleavings. Key contributions include: A modular search framework with four core components: HNAS Engine (search algorithm), Evaluator (dataset selection), Aggregator (candidate synthesis), and Extrapolator (scaling to large models) .

**Strengths:**

1.Composer addresses a critical gap in hybrid architecture design by automating the exploration of vast design spaces (e.g., 4 billion configurations for 32-layer models) . The integration of modular components (e.g., clustering-based aggregation  and synthetic dataset evaluation) ensures both efficiency and robustness.

2.The proposed stacking and stretching techniques enable seamless scaling of small architectures to large models. For example, stretching 16-layer configurations preserves global interleaving patterns, outperforming stacking in larger models .

**Weaknesses:**

1. While MAD enables efficient search, its token-manipulation tasks may not fully represent real-world complexity. The paper lacks a comparison with other proxy datasets.
2. Results are validated up to 3B parameters, but the framework’s applicability to larger models (e.g., 7B+) remains untested.
3. The paper does not fully analyze the final searched structures, e.g., why specific interleavings (e.g., 1:2 Attention/MLP ratios) outperform others.

**Questions:**

Please see the weaknesses.

---

> ### Author Response · Authors · 2025-11-20
> **Answer to Reviewer naoJ's Questions/Weaknesses:**
>
> ## Answer to Q1:
> To the best of our knowledge, MAD and BabiStories are the only two proxy datasets available, both of which we use in our experimental analysis (Section 3.2). MAD does not cite any prior proxy datasets as well. The use of proxy datasets for rapid evaluation is a relatively new research topic (MAD was introduced at ICML’24, BabiStories was introduced in 2025 by Meta Research). Through empirical analysis between MAD, BabiStories, and DCLM (a widely used text corpus), we find that MAD works well for evaluating the efficacy of LLM architectures when conducting small-scale search (Section 3.2, Figure 2 and Appendix E.6, Figure 14). Composer’s modular design enables rapid evaluation of new proxy datasets that the community proposes. If there are suggestions on other proxy datasets, we are happy to investigate.
>
> ## Answer to Q2:
> We collected pre-training results at 8B scale with a compute budget of 4e20 FLOPs. The results on the 6 downstream tasks shows that our Composite LLMs continue to outperform baseline Llama models. Note that the 8B architecture is produced by extrapolating via stacking (6-layer search) and stretching (16-layer search), following the same methodology to produce the 350M-3B models.
>
> | Size | Model | Loss | Arc C.|  Arc E. |  Hella. |  Wino. |  SciQ |  PIQA | Avg. |
> |-------|--------|----|----|----|----|----|----|----|----|
> | 8B | Llama 3.2 | 3.03 | 42.2 | 70.9 | 70.2 | 66.0 | 90.5 | 76.5 | 69.3 |
> | 8B | Composite - Stacked | 2.95  | 42.7 | 71.5 | **70.6** | 64.8 | 91 | 77.4 | 69.7 |
> | 8B | Composite - Stretched | **2.93** | **43.6** | **71.8** | 70.4 | **66.5** | **91.3** | **78.5** | **70.4** |
>
> We are also exploring another methodology for extrapolating our small discovered LLMs to larger sizes (e.g., 8B). Currently, we either stretch OR stack to extrapolate to larger scales. We see that stretching outperforms stacking at 3B model size, as it maintains a quality interleaving pattern and ratio of the computational primitives. However, continuously stretching small discovered LLMs to larger sizes may degrade model quality, as large chunks of the LLM’s layers will be dominated by a single computational primitive (e.g., all GQA block for several layers in a row). Such models would have poor interleaving of computational primitives.
>
> There are two methodologies we propose to overcome this. We could increase the number of layers and trials in our search. However, this comes with a tradeoff of increased search cost. Hence, we propose an alternative strategy that combines both stretching and stacking. This methodology will create a high-quality 3B model via stretching that maintains a good interleaving pattern and ratio of primitives, and then, by stacking into 8B, maintain quality interleaving and ratio of primitives without large chunks of the LLM being dominated by a single primitive. We will add our findings after we pre-train the hybrid LLMs produced with this extrapolation methodology.
>
> ## Answer to Q3:
> To the best of our knowledge, there is no reason to expect the standard 1:1 stacked Attention/MLP to be optimal, as also noted in prior work [Sandwich Transformer, ACL’20](https://aclanthology.org/2020.acl-main.270/).  There is numerous work in the post-training pruning literature showing that attention in later layers are redundant and can be pruned during inference without impacting accuracy (CHAI [ICML’24](https://arxiv.org/abs/2403.08058), How Much Does Attention Actually Attend? [EMNLP’22](https://arxiv.org/abs/2211.03495), What matters in Transformers? Not all attention is needed [arxiv’24](https://arxiv.org/abs/2406.15786)). Composer shows the optimal structures can be discovered automatically and optimized before pre-training, resulting in higher accuracy models and efficient training.

---

### Official Review · Reviewer_romG · 2025-10-31

**Soundness:** 4
**Presentation:** 3
**Contribution:** 3
**Rating:** 8
**Confidence:** 4

**Summary:**

This paper introduces a framework for optimising the combination of different computational primitives (like attention or MLP blocks) in a neural network, towards hybrid architectures. It uses scaling strategies beyond scaling laws to search on small models that can later be scaled up. They focus on finding architectures for pretraining and ablate over choices for search, evaluation, aggregation and extrapolation in a comprehensive manner. They report consistent improvements in their found hybrid architectures on LLM benchmarks, including improved efficiency and downstream performance.

**Strengths:**

The experimental methodology is strong, with different components tested in isolation, and the claims of the paper appear well evidenced.
The paper is presented well overall, notably figure 1 and the 4 key design questions neatly structure the paper. This paper includes a strong ablation study across several dimensions of the proposed framework, which is presented in a clear structure with highlighted observations/key findings and further discussion.
The overall contribution of this paper is to provide a well designed framework for searching and scaling transformer architectures for LLM pretraining that can be useful for the broader ICLR community.

**Weaknesses:**

While the presentation of the method and experiments are good, the main paper is missing a Related Work section, which would significantly help in clarifying the surrounding literature and the contribution of this work, especially compared to the STAR method. Moving this in part from the appendix to the main paper would improve the clarity here.

More ablation could be done on the clustering method that is claimed to smooth out noise. Only Figure 6 (bottom) and the short discussion on it go into the effect of this clustering approach, which seems to be a core reason for the strong performances achieved. I also found the clustering explanation the hardest section to understand. Perhaps a visual aid could clarify it, e.g. a diagram of N_0 vs N_1 vs N_all or similar.

Issues in text and tables:
* L265-267: I don’t see how both One-Shot methods have lower search cost than End-Incremental. Figure 2 (left) suggests it is the same for 6-layer and more costly for 16-layer.
* L320-321: What is meant by better capturing global information here?
* Key results 1 and 2: Are these results computed across both the Composite - Stacked and Composite - Stretched models? A clarifiction is needed here
* Table 6, final row reports an incorrect Avg. value for the Composite - Stretched model. It should be 62.0 instead of 66.9. I would point out to make sure the percentage claims in the abstract and elsewhere are not based on this mistake.
* C.6 Baselines: If I understand correctly, the Striped attention should have its 1:4 ratio as 5x(1A+4M) but it says 5x(1A +2M). Same for 1:8.

**Questions:**

- To quantify the robustness of the Composer framework for other types of pretraining data, it would be good to have an experiment that swaps DCLM for another corpus. Would the models found on different pretraining data also outperform Llama 3.2?
- How many trials are performed of Bayesian Optimisation for the One-Shot 6-layer setup? The search space is 2^6 = 64 and it seems as 100 trials were performed for the 16-layer setup. Can a comprehensive evaluation of all architectures in the search space be done with an analysis how well the best can be found?
- What task is the search accuracy evaluated on for Figure 1 (right)?
- What are the details on the Bayesian Optimisation approach for One-Shot searches? What kernel/acquisition function/hyperparameters? There seems to be no details on this.

---

> ### Author Response · Authors · 2025-11-20
> **Answer to Reviewer romG's Questions**
>
> ## Answer to Q1:
> Thank you for your suggestion. We are actively running experiments to validate that our Composite models outperform Llama 3.2 if pre-trained on a different corpus, using the c4 training corpus.
>
> ## Answer to Q2:
> For a 6 block, we search the entire space (64 trials), since it is below the trial threshold (100 trials). Unfortunately, a comprehensive evaluation of all architectures is quite expensive: it requires pre-training 64 hybrid LLM architectures from 350M - 3B model parameter sizes. We estimate this would take >30K GPU-hours, which we cannot complete during the rebuttal period.
>
> To understand how the distribution of candidate hybrid architectures performs when pre-trained at scale, we ranked the candidate architectures based on their small-scale search performance on the MAD dataset, and then pre-trained the p0-p100 architectures (i.e., p0: worst performing model candidate, p100: best performing model) at 1B model size (Section 4.4, Figure 6). The Spearman rank correlation between small-scale 6-layer search and 1B scale is 0.97 on average across compute budgets. Figure 6 shows that our Composite LLM outperforms the p0-p100 models at scale, indicating that Composer can find the best quality architecture during its small-scale search.
>
> ## Answer to Q3:
> Figure 1 results are collected with the MAD benchmark. We will clarify this in our paper.
>
> ## Answer to Q4:
> Our Bayesian optimization method is built upon Ax / BoTorch SingleTaskGP model with an RBF Kernel and dimension-scaled priors, and qLogNEI acquisition function for single objective optimization (details can be found at: “[Ax: a platform for adaptive experimentation](https://openreview.net/pdf?id=U1f6wHtG1g)” [AutoML’25]). All of the searches are done using the same algorithm. We will include these details in the paper.

---

> > ### Author Response · Authors · 2025-12-01
> > **Update: Answer to Q1 - Pretraining on different corpus**
> >
> > We have completed our runs with the c4 dataset as the pretraining corpus. Here are the results:
> >
> > | Model | Dataset | Size | Budget | Train Loss | Val Loss |
> > | --- | --- | --- | --- | --- | --- |
> > | Llama | C4 | 1B | 4.00E+20 | 2.732 | 2.689 |
> > | Composer - Stretched | C4 | 1B | 4.00E+20 | 2.655 | **2.643** |
> > | Composer - Stacked | C4 | 1B | 4.00E+20 | **2.575** | 2.648 |
> >
> > The results show that models found with Composer outperforms Llama 3.2 on a different pre-training corpus as well, validating the proxy tasks used are good proxies.

---

> ### Author Response · Authors · 2025-11-20
> **Answer to Reviewer romG's Weaknesses**
>
> ## Answer to W1:
> Thank you for your suggestion. We will move the Related Work section from the appendix to the main manuscript to distill our contributions compared to prior work.
>
> ## Answer to W2:
> We are actively working on a visual aid to include in our appendix that intuitively captures the methodology of $N_c$ clustering. We will include that visual aid here shortly for feedback as well. Please refer to our answer to Reviewer USsZ (section "Answer to W4") for a further in-depth theoretical explanation of why $N_0$ clustering performs well. We will include this explanation in our paper.
>
> ## Answer to W3 (clarifications of text/tables):
>
> ### L265-267
> We clarify that the 16-layer One-Shot has a longer search cost than End-Incremental. Meanwhile, both 6-layer One-Shot and End-Incremental have the same search cost. The search cost predominantly depends on the number of candidate hybrid architectures evaluated during the search. Both End-Incremental and 6-layer search evaluate 64 candidate architectures, and hence they have the same search cost:
> - 6-layer One-Shot: there are 2^6 potential architectures (2 computational primitive options for each layer index)
> - End-Incremental: We progressively increase the architecture depth by n layers at each step, fixing computational primitives for the previous layers and searching only the last n. We set n=2. Hence, during each iteration, there are 2^2 potential architectures to evaluate. We complete our search when the depth of the architecture is 32 layers. Therefore, there are 16 iterations of search, resulting in 64 architectures evaluated with this search methodology.
>
> 16-layer One-Shot has a greater search cost than 6-layer One-Shot and End-Incremental, since we evaluate more than 64 candidate architectures (we cap our search at 100 trials) with this search methodology.
>
> ### L320-321:
> During Composer’s small-scale search, it learns locations to transition from one computational primitive to another (e.g., attention to MLP). These transitions are important for the hybrid LLM to retain and propagate information and signals/gradients across layers to capture more complex dependencies. Stretching as an extrapolation technique naturally preserves these transition points at larger scale. We will clarify this wording in the paper.
>
> ### Key results 1&2:
> These results are computed across both the Composite models. We will clarify this in our paper.
>
> ### Table 6 and C.6 Baselines:
> Thank you for catching these details in our appendix! We will fix Table 6 accordingly with the correct average value of 62.0. We note that the percentage claims in our abstract, main paper, and Appendix E.4 are still correct and not based on this mistake. Moreover, your understanding for the striped attention baselines in C.6 is correct, we will fix this accordingly.

---

> > ### Comment · Reviewer_romG · 2025-11-27
> > **Answer to rebuttal**
> >
> > I thank the authors for their answers and their clarifications regarding my questions. I would also appreciate if the paper could be updated already accordingly.
> > Furthermore, I am happy to keep my score.

---

### Official Review · Reviewer_ziXT · 2025-11-01

**Soundness:** 3
**Presentation:** 3
**Contribution:** 2
**Rating:** 6
**Confidence:** 4

**Summary:**

This paper introduces Composer, a modular framework for automatically discovering hybrid large language model architectures that outperform standard Transformers. While standard transformers are composed by a uniform 1:1 ratio of alternating attention and mlp layers, the authors found that a 1:2 ratio is a/ more efficient to compute and b/ (marginally) better performing on multiple choice benchmarks

**Strengths:**

- very relevant topic, other works also propose that the standard 1:1 ratio might be suboptimal
- LLM's can be more efficient (computation) and potentially more performant.
- solid framework and experimental design

**Weaknesses:**

W1 unfortunately only multiple choice downstream benchmarks were tested. you propose to alter 'the core' of current LLM's, it is pretty unclear whether this architecture change will also perform on disciplines as long context, coding, math, extended reasoning, or massive multi-linguality. at least for the final 3b models you should do one more ablation in these directions, baseline vs your 1:2 mix, to also validate this finding still holds.

W2 isn't it correct that your 1b MLP has a different scale-up factors? shouldn't that skew search results?

W3 will you publish your code before the conference?

**Questions:**

please address above's weaknesses

---

> ### Author Response · Authors · 2025-11-20
> **Answer to Reviewer ziXT's Questions/Weaknesses:**
>
> ## Answer to Q1:
> We are working on fine-tuning our models for evaluating on reasoning tasks such as GSM8K and Math. We will get back with the results before the rebuttal deadline. We have few-shot evaluation results on Math, GSM8K and MMLU benchmarks without fine-tuning for your reference in Table A. While the results are low for Llama 3.2 and our Composite models (we are actively fine-tuning to improve performance), the results indicate that our Composite LLMs slightly outperform Llama 3.2 on math and reasoning tasks.
>
> |         Model        |  Math | GSM8K |  MMLU  |
> |:--------------------:|:-----:|:-----:|:------:|
> | Llama-3.2            | 2.50% | 1.67% | 31.06% |
> | Composite: Stacked   | 2.84% | 2.07% | 31.33% |
> | Composite: Stretched | 2.60% | 1.82% | 31.48% |
> Table A. Math, GSM8K, MMLU results for Llama3.2 and Composite models. Model size is 1B and compute budget is 4e20. The models are only pretrained on DCLM without post-training. MMLU score is with continuation generation.
>
> In addition to these results and fine-tuning our current Composite architectures, we note that to improve performance on the suggested disciplines, we would include other computational primitives in our search (e.g., sliding window attention), as previous works show that LLM architectures with such primitives are better suited for long-context and extended reasoning (InfLLM NeurIPS’24, CAB ICML’23) . In this work, we narrow the scope of our primitives to GQA and SwiGLU layers to first build a robust HNAS framework that can be extended for future LLM architecture design. While we have not included sliding window attention in our search, we’ve conducted experiments with other computational primitives (e.g., state space models like Mamba, COLM’24 ), and our results indicate that Composer can effectively navigate the increased search space and discover new hybrid architectures with better model quality.
>
> Moreover, the dataset we use for HNAS’s Evaluator is not designed to optimize for reasoning or long context tasks. Rather, MAD’s proxy benchmarks are designed for common language tasks. To our knowledge, no small-scale datasets exist that target reasoning/long context tasks, which ideally we would use to discover hybrid LLMs targeting such tasks. As the use of proxy datasets for quick reliable signals is a research avenue that has recently gained traction (MAD appeared in ICML’24, the TinyStories/BabiStories datasets appeared in arxiv’23/ICLR’25), Composer’s modular design enables future studies to rapidly evaluate the efficacy of such new datasets targeting other LLM tasks.
>
> ## Answer to Q2:
> Depending on the model size we scale to, the scale-up factor is different (for 350M to 3B). All the baselines are scaled up to the same number of total model parameters and trained with the same compute budgets or same token budgets (for results in Table 2) making the evaluation fair.
>
> ## Answer to Q3:
> Yes, we plan to open-source the framework code along with scripts to reproduce the results upon acceptance.

---

### Official Review · Reviewer_USsZ · 2025-11-03

**Soundness:** 3
**Presentation:** 3
**Contribution:** 3
**Rating:** 6
**Confidence:** 3

**Summary:**

Overall, this work addresses important issues in NAS and demonstrates a fairly high level of research, providing extensive experimental results through numerous experiments. The writing is relatively clear, and the technical details are complete, though it lacks deeper theoretical analysis.

If the theoretical analysis and intuitive understanding of the experimental results and proposed methods could be strengthened, I would be willing to raise the score.

**Strengths:**

This article proposes a search framework for hybrid neural architecture design, conducts a variety of experiments, and achieves significant performance improvements.

**Weaknesses:**

1. The fourth key issue raised by the authors is important among the four issues. This paper introduces the approach of neural architecture extrapolators and empirically demonstrates its effectiveness, but the rationale behind this extrapolation method lacks analysis and justification. For how to extrapolate from smaller-scale networks to larger-scale networks, especially the possible explanation, it is suggested to refer to the following literature for a theoretical explanation.

MathNAS: If Blocks Have a Role in Mathematical Architecture Design, Sihai Zhang et al, NeurIPS, 2023

2. The first key issue raised in this paper is not that important, unless this small-scale model architecture search differs significantly from other small-scale architecture searches when scaling up, which is not discussed in this paper.

3. The second key issue proposed in this paper is also difficult to explain. The paper only completes experimental comparisons and does not, and I believe cannot, provide insightful explanations as to why this dataset is used rather than that one.

4. Perhaps the third key issue raised in this paper is the true contribution of the work. Using the idea of clustering to form the final network seems interesting. However, the related content included in the appendix does not discuss why clustering works. It is suggested that the authors provide an explanation for this.

**Questions:**

1. I feel that the main focus of this work is not clearly highlighted. I suggest the author clarify what the central focus of this work is. Is it the discovered Composite architecture or the proposed Composer framework?

2, Although four key questions are raised, they are not well answered. I recommend that the author provide deeper and more insightful conclusions for one or two of these questions. This might be better.

3. The author claims, 'During search, the HNAS Evaluator trains and evaluates candidate hybrid LLMs with a small dataset to provide fast, reliable signals on the potential quality of the architecture at scale.' Please explain why a small dataset can provide reliable quality signals for large-scale models. How dependent is this signal on the dataset? I am very curious about this.

---

> ### Author Response · Authors · 2025-11-20
> **Answers to Reviewer USsZ's Questions**
>
> ## Answer to Q1:
> The contribution of our work is two-fold.
> 1. We are the first study to design a robust search framework in a principled manner that enables rapid discovery of new hybrid LLM architectures. As the community proposes new building blocks (e.g., Mamba COLM’24, Hyena ICML’23, Gated DeltaNet ICLR’25), the design space for new hybrid LLMs exponentially grows. Currently, the model architecture design process is manual and based on intuition — no systematic framework exists today to enable automatic, efficient discovery of hybrid LLM architectures that perform well at scale. We are the first to systematically design an HNAS framework by studying four key design questions (outlined in Section 1, Introduction). Of the design questions, 2-4 are where we offer many contributions. We distill our contributions for each of these design questions in our answer to the reviewer's question #3 and weaknesses #1 and #4.
> 2. We then validate the efficacy of our finalized framework by showcasing two architectures  Composer discovers, which improve the quality and efficiency of the LLM compared to Llama 3.2 and other prior architectures.
>
> ## Answer to Q2:
> Please refer to our answers to Q1, Q3, W1, and W4.
>
> ## Answer to Q3:
> We respond to this question by distilling two subquestions. The first question is: why are the proxy token-manipulation tasks found in MAD good evaluators for at-scale LLM performance? This question is challenging to answer and a very relevant open research question. The use of proxy datasets for rapid evaluation is a recently proposed direction (MAD appeared in ICML’24, the TinyStories/BabiStories datasets appeared in arxiv’23/ICLR’25), and our empirical results further showcase their efficacy in enabling Composer to discover its Composite LLMs efficiently. However, further theoretical justification for why these datasets work well is left for future work.
>
> The second question is: why do small datasets provide better signals than the large datasets we pre-train with? We first note that in Figure 2 (right plot), we show that with a downsampled version of the large DCLM dataset, we do in fact discover hybrid LLMs that outperform Llama 3.2 (labeled “Large-scale DCLM”). However, to discover quality LLMs with this dataset, we had to search with 150M parameter model sizes. We found that conducting search using this large dataset with 1-2M parameter model sizes produced worse-performing LLMs (labeled “Small-scale DCLM”). We hypothesize that this is because the vocabulary size of the dataset (128K) is too large and complex for small LLMs when conducting small-scale search. The performance of the small LLMs with a large dataset is not indicative of at-scale performance and is simply noise, hindering Composer from discovering quality LLM architectures.
>
> Meanwhile, the vocab size of MAD (16-256) is more manageable for the small 1-2M models we search with, enabling Composer to discover high-performing architectures. One of our design constraints was to create an efficient HNAS framework, rendering search at 150M parameter sizes too expensive. Under this constraint, we found that small datasets provide the cleanest, most informative signals for architecture discovery. We suspect that as the community introduces more high-quality small datasets for rapid evaluation, leveraging them in our modular design would enable Composer to continue to discover high-quality hybrid LLMs.
>
> Hence, a core finding from our study of datasets with our second design question is not necessarily that small datasets are better than large ones for discovering hybrid LLMs, but rather both can be effective and choosing between them ultimately depends on the search cost one is willing to pay. Utilizing smaller datasets when searching with small models is more effective for discovering quality LLMs with low search cost.

---

> ### Author Response · Authors · 2025-11-20
> **Answer to Reviewer USsZ's Weaknesses**
>
> ## Answer to W1:
> Conventional wisdom for model architectures is to stack core blocks to a desired size. However, to the best of our knowledge, we are unaware of theoretical proofs in prior works that guarantee the optimality of stacking as an extrapolation technique. Moreover, there is no reason to expect the standard 1:1 stacked transformer architecture to be optimal, to the best of our knowledge, as also noted in prior work [Sandwich Transformer, ACL’20].
> Hence, following conventional wisdom, we used Composer to discover new core modeling blocks and stacked these blocks to extrapolate to larger sizes. However, we also considered stretching as an extrapolation technique. Our intuitive reasoning for stretching is that it preserves the searched structures better than stacking: it maintains the ratio and interleaving pattern of computational primitives. This may be why our stretched Composite LLM slightly outperforms our stacked one.
>
> We do not claim that stacking or stretching methods are optimal extrapolation methods. Our work showed empirically that stretching can work better in some cases. We hope this provides a better intuitive understanding. We defer theoretical investigation to future work.
>
> We appreciate the reviewer's suggestion to look at MathNAS. Similar to Composer, MathNAS uses an incremental search process (Equation 2) to build up its network to the desired size; it does not leverage any extrapolation techniques. Hence, the analysis offered in Appendix B.1 (https://arxiv.org/pdf/2311.04943) justifies why Composer’s iterative search methodologies are theoretically sound, however, we cannot use their analysis to theoretically justify our extrapolation techniques. We will cite MathNAS in our paper for theoretical justification of our search methodologies.
>
> ## Answer to W3:
> Please refer to our answer for question 3.
>
> ## Answer to W4:
> The following explanation provides further intuitive reasoning as to why $N_0$ clustering works well, which aggregates top-performing architectures by selecting the most frequent block at each layer independently.
>
> Consider each architecture as a sequence of blocks across layer indices. The search process returns a set of top-performing architectures, which we filter based on validation accuracy with MAD. We interpret this collection as a representative Monte Carlo sample from a distribution over high-performing designs. This distribution is implicitly shaped by the architecture search process — in our case, Bayesian optimization (BO). The BO surrogate model evaluates full architectural configurations and captures interactions between blocks at different layers, learning correlations between structural choices and validation performance. As a result, the search does not sample randomly, but instead includes architectures that exhibit useful inter-layer dependencies.
>
> Given this, we can define an empirical frequency for how often each block appears at a given layer across the sampled architectures. This frequency estimates how likely that block is to appear at that layer in a high-performing design. $N_0$ clustering selects, for each layer, the block that appears most frequently. The final architecture is then assembled by independently choosing the most common block at each layer across the top-performing candidates.
>
> Selecting the block with the highest marginal frequency at each layer is equivalent to computing the architecture that maximizes the product of these marginal frequencies across all layers. In other words, $N_0$ clustering selects a configuration that is most probable under the assumption that each layer’s block choice is independent of the others. This corresponds to a Naive Bayes-style estimator — a well-known technique in probabilistic modeling — which maximizes the product of marginal likelihoods for individual variables while ignoring conditionals.
>
> While this formulation ignores explicit inter-layer dependencies during aggregation, we argue this is appropriate because such dependencies are already accounted for during the search phase. That is, the BO search process acts as a dependency-aware sampler to construct the set of top-performing architectures. $N_0$ clustering aggregates across a structurally pre-filtered space, smoothing out noisy or overfitting samples while preserving dominant patterns.
>
> Under this framing, $N_0$ clustering is a statistically consistent estimator of the most likely block sequence under a marginal model of high-performing architectures. By prioritizing frequent blocks and avoiding conditional noise or overfitting, $N_0$ clustering yields architectures that generalize well when extrapolated to larger scales.
>
> We also add explicit inter-layer dependencies in our clustering techniques with $N_1$ and $N_i$ clustering. We empirically evaluate them against $N_0$ clustering and find that they perform worse (Figure 13 in Appendix E.5), showing empirical evidence that $N_0$ clustering is a sound clustering technique.

---

### Meta-Review · Area_Chair_AMdW · 2026-01-12

**Summary:**

This paper addresses the important problem of architecture design, and shows that it is still relevant in the age of scaling, by showing that searching over architectures can give architectures that perform better, even as compute and data is scaled up. The reviewers raised several questions about whether the empirical results in the paper would hold in general (e.g. scaling beyond 3B). This is always a concern with empirical work, and the authors have provided additional evidence. Given that these confirm existing claims, rather than fix flaws in the paper, I am inclined to accept this paper.

**Reviewer Concerns:**

Reviewers had several concerns, e.g. the generality of the framework to incorporate other architectural structures, the scalability of architectures beyond 3B, and the applicability to other datasets. The authors have performed additional experiments, showing similar results

**Reviewer Scores:**

Overall, it is hard to determine how scores would have changed, although some of the negative reviews may have increased had they engaged. The most positive reviewer did engage with the rebuttal, and kept his score.

---

### Decision · Program_Chairs · 2026-01-26

Accept (Poster)